# Stabilization of membrane topologies by proteinaceous remorin scaffolds

Chao Su[1], Marta Rodriguez-Franco [1], Beatrice Lace[1], Nils Nebel [1], Casandra Hernandez-Reyes[1,2], Pengbo Liang[1], Eija Schulze[1], Evgeny V. Mymrikov [2,3], Nikolas M. Gross [2,3,4], Julian Knerr [5], Hong Wang [2,5], Lina Siukstaite[1,6], Jean Keller [7], Cyril Libourel [7], Alexandra A. M. Fischer [2,4,6,8], Katharina E. Gabor[9], Eric Mark[10], Claudia Popp[11], Carola Hunte [2,3,6], Wilfried Weber [2,6,8], Petra Wendler [10], Thomas Stanislas[9], Pierre-Marc Delaux[7], Oliver Einsle[12], Robert Grosse [2,5], Winfried Römer [1,2,6] & Thomas Ott [1,2] ✉

In plants, the topological organization of membranes has mainly been attributed to the cell wall and the cytoskeleton. Additionally, few proteins, such as plant-specific remorins have been shown to function as protein and lipid organizers. Root nodule symbiosis requires continuous membrane re-arrangements, with bacteria being finally released from infection threads into membrane-confined symbiosomes. We found that mutations in the symbiosis-specific *SYMREM1* gene result in highly disorganized perimicrobial membranes. AlphaFold modelling and biochemical analyses reveal that SYMREM1 oligomerizes into antiparallel dimers and may form a higher-order membrane scaffolding structure. This was experimentally confirmed when expressing this and other remorins in wall-less protoplasts is sufficient where they significantly alter and stabilize *de novo* membrane topologies ranging from membrane blebs to long membrane tubes with a central actin filament. Reciprocally, mechanically induced membrane indentations were equally stabilized by SYMREM1. Taken together we describe a plant-specific mechanism that allows the stabilization of large-scale membrane conformations independent of the cell wall.

Intracellular colonization of host cells represents a characteristic feature of mutualistic associations like the root nodule symbiosis (RNS) and the arbuscular mycorrhiza symbiosis (AMS) that occur between a host plant and soil-borne bacteria, such as rhizobia, or Glomeromycotean fungi, respectively. Molecularly, RNS-related infection is tightly controlled and requires the perception of strain-specific rhizobial lipochitooligosaccharides (Nod factors) by host LysM-type receptor-like kinases[1-3] and of rhizobial exopolysaccharides[4]. Nod

[1]Faculty of Biology, University of Freiburg, 79104 Freiburg, Germany. [2]CIBSS – Centre of Integrative Biological Signalling Studies, University of Freiburg, 79104 Freiburg, Germany. [3]Institute for Biochemistry and Molecular Biology, ZBMZ, Faculty of Medicine, University of Freiburg, 79104 Freiburg, Germany. [4]Spemann Graduate School of Biology and Medicine (SGBM), University of Freiburg, 79104 Freiburg, Germany. [5]Institute of Pharmacology, Medical Faculty, University of Freiburg, 79104 Freiburg, Germany. [6]BIOSS – Centre for Biological Signalling Studies, University of Freiburg, 79104 Freiburg, Germany. [7]Laboratoire de Recherche en Sciences Végétales (LRSV), Université de Toulouse, CNRS, UPS, INP Toulouse, Castanet Tolosan, France. [8]Division of Synthetic Biology, Faculty of Biology, University of Freiburg, 79104 Freiburg, Germany. [9]Center for Plant Molecular Biology (ZMBP), University of Tübingen, 72076 Tübingen, Germany. [10]Institute of Biochemistry and Biology, Department of Biochemistry, University of Potsdam, 14476 Potsdam-Golm, Germany. [11]Ludwig-Maximilians-University (LMU) Munich, Institute of Genetics, 82152 Martinsried, Germany. [12]Institute of Biochemistry, Faculty of Chemistry, University of Freiburg, 79104 Freiburg, Germany. ✉e-mail: Thomas.Ott@biologie.uni-freiburg.de

factor perception by the host and heteromeric receptor complex assembly trigger a symbiotic signaling cascade that results in peri-nuclear calcium spiking[5]. This calcium signature is, in turn, decoded by the calcium-calmodulin-dependent kinase CCaMK/DMI3 and the transcriptional activator CYCLOPS/IPD3[6,7]. Upon phosphorylation of CYCLOPS/IPD3 by CCaMK/DMI3, RNS-specific gene expression is triggered by the activation of specific transcription factors such as NODULE INCEPTION (NIN)[8,9]. Consequently, mutations in these genes result in the inability of the host to maintain intracellular infections and bacterial release.

Intruding rhizobia mostly infect legume hosts such as *Medicago truncatula* and *Lotus japonicus* via young, growing root hairs that swell and later curl around surface attached rhizobia to entrap them just below the root hair tip. This process is driven by cellular repolarization of the actin and microtubule cytoskeleton[10–13] and involves, among others, the actin polymerizing formin protein SYFO1[14]. Root hair curling results in an entrapment of the symbiont in a so-called infection chamber[15]. A re-polarization of the cell towards the infection chamber leads to the targeted secretion of proteins and membrane constituents, which enables the formation of a membrane-surrounded tunnel called the 'infection thread' (IT) that emerges from the infection chamber, transcellularly progresses though the root cortex and finally branches inside the nodule primordium[16,17]. Within these primordia and later in indeterminate nodules of *M. truncatula*, rhizobia are continuously released from bulges of nodular ITs (infection droplets) into these infected cells. Upon release, rhizobia differentiate into nitrogen-fixing bacteroids that remain encapsulated by the host-derived symbiosome/peribacteroid membrane[18,19]. It should be noted that pre-infection membrane invaginations and constitutive IT stabilization during RNS are most likely host-driven processes. In addition, initial membrane invaginations, young IT segments around the growing tip, infection droplets as precursors of bacterial release sites, and symbiosome membranes encapsulating the nitrogen-fixing symbiont are devoid of a rigid cell wall that could provide structural support to these sites[20,21]. Although this is crucial to the success of the infection, our understanding of symbiotic membrane stabilization in the absence of a primary cell wall is rather sparse.

Such roles could alternatively be maintained by oligomeric scaffold proteins like clathrins as described during endocytosis[22,23] or Bar-domain proteins like amphisin or BIN1 in human cells[24]. During RNS, the scaffold SYMREM1, a member of the plant-specific remorin family[25], recruits and stabilizes the symbiotic receptor LYK3 in membrane nanodomains[26]. This function might be explained by remorin-induced alterations in membrane fluidity[27,28], higher order protein oligomerization[29–31] or maintenance of membrane-associated and phase-separated condensates[32]. Receptor scaffolding additionally relies on the Flotillin protein FLOT4[33]. While *FLOT4* is already expressed prior to rhizobial infection, *SYMREM1* expression is exclusively triggered in the presence of the symbiont[26,34,35]. Furthermore, SYMREM1 localization to membrane nanodomains, but not to the plasma membrane per se, depends on FLOT4[26]. Like all other remorins, SYMREM1 is comprised of a conserved alpha-helical C-terminal region, while the intrinsically disordered N-terminal region (IDR) of SYMREM1 is highly variable in sequence and conformation[36]. Membrane association of remorins is mediated by the Remorin C-terminal Anchor (RemCA) and, in most cases, assisted by palmitoylation[36,37]. Furthermore, it has been shown that remorins oligomerize at the plasma membrane in planta[38] and can form higher order filamentous structures or protein lattices in vitro[31].

Since remorins can alter membrane fluidity[28] and *symrem1* mutants largely fail to release rhizobia into nodule cells[35], these proteins may have greater impact on membrane topology than currently envisioned. Furthermore, SYMREM1 accumulates predominantly on cell wall-devoid symbiotic membranes such as IT tips[26], nodular infection droplets that precede bacterial release and the symbiosome

membrane, which surrounds the released and nitrogen-fixing rhizobia inside the nodule[35]. Therefore, we investigated putative roles of SYMREM1 in membrane dynamics and shape.

## Results

### SYMREM1 controls symbiotic membrane topologies

To visualize symbiotic membranes in great detail and to examine their precise morphology, we labeled phosphatidylserine (PS), a central phospholipid of biological membranes, by expressing a LactC2 biosensor[39]. This allowed clear imaging of membrane patterns at ITs, infection droplets and symbiosome membranes (Supplementary Fig. 1a–f). As previously shown and recapitulated here, all three membrane sites were also targeted by SYMREM1 (Fig. 1a–f; Supplementary Fig. 1g–i)[35]. While ITs and infection droplet structures were mostly filled with rhizobia, we frequently observed additional empty tube-like membrane structures on enlarged ITs associated with infection droplets in nodule cortex cells of wild-type (WT) plants (Fig. 1g). In line with this accumulation of SYMREM1 at ITs, on infection droplets and on symbiosome membranes, and in contrast to WT plants (Supplementary Fig. 2a, d), *symrem1* mutants are greatly impaired in releasing bacteria to cells of the inner nodule cortex and exhibited bulky ITs, as revealed by light and electron microscopy (Supplementary Fig. 2b–c, e–f)[35]. To assess possible alterations in membrane topology, we also expressed the PS-biosensor in the *symrem1-1* mutant. In contrast to WT plants, ITs observed in the *symrem1* mutant did not display signs of membrane tubulation (Fig. 1h). While differentiated bacteroids with a tightly aligned symbiosome membrane were observed in WT nodules (Fig. 1f, k; Supplementary Fig. 2g, j), symbiosome membranes were either loosely aligned with bacterial shapes (Fig. 1i, l–m; Supplementary Fig. 2h–i, k–l) or released as empty membrane spheres (Fig. 1j, m, n). While loosely attached symbiosome membranes were observed in all *symrem1* mutant nodules, only some nodule cells contained empty membrane spheres. These structures rather resembled symbiosome membranes than tonoplasts as they contained PS (Fig. 1j), a lipid that does not occur on vacuolar membranes[40].

So far, SYMREM1 has been described as a molecular scaffold that interacts with and stabilizes the entry receptor LYK3 at the plasma membrane[26,35]. While LYK3 has been localized to ITs[26,33], no persistent accumulation on the infection droplet nor the symbiosome membrane has been observed[41]. This is in contrast to SYMREM1 that remained and even accumulated on these unwalled symbiotic membranes (Fig. 1d–f; Supplementary Fig. 1g–i) implying that SYMREM1 serves additional functions. Given the lack of stabilized tubular outgrowths from ITs close to infection droplets, our findings that symbiosome membranes appeared less supported and consequently loosely organized around the bacteroid and the occurrence of empty membrane spheres in *symrem1* mutants, we hypothesized that SYMREM1 functions in maintaining distinct topologies of unwalled membranes (summarized in Supplementary Fig. 3).

### SYMREM1 functions as a structural membrane scaffold

The hypothesis of SYMREM1 being a structural membrane scaffold was further supported by the fact that other members of the remorin family have been shown to form higher order proteinaceous lattices and/or filamentous structures[30,31,42,43] in vitro. We also confirmed such filamentous structures for purified, recombinant SYMREM1 using TEM analysis followed by 2D classification of manually picked filamentous segments. Here, we detected auto-assembled amorphous protein filaments that were partially branched or scrambled (Fig. 2a). Systematic inspection of 389 filament fragments revealed an average width between 84 and 125 Å for the filamentous particles with a few class averages showing helical features such as twists (Fig. 2b). Besides this, we mostly observed irregular filaments and protein bodies (Fig. 2a) that may represent filament seeds. Analytical gel-filtration of

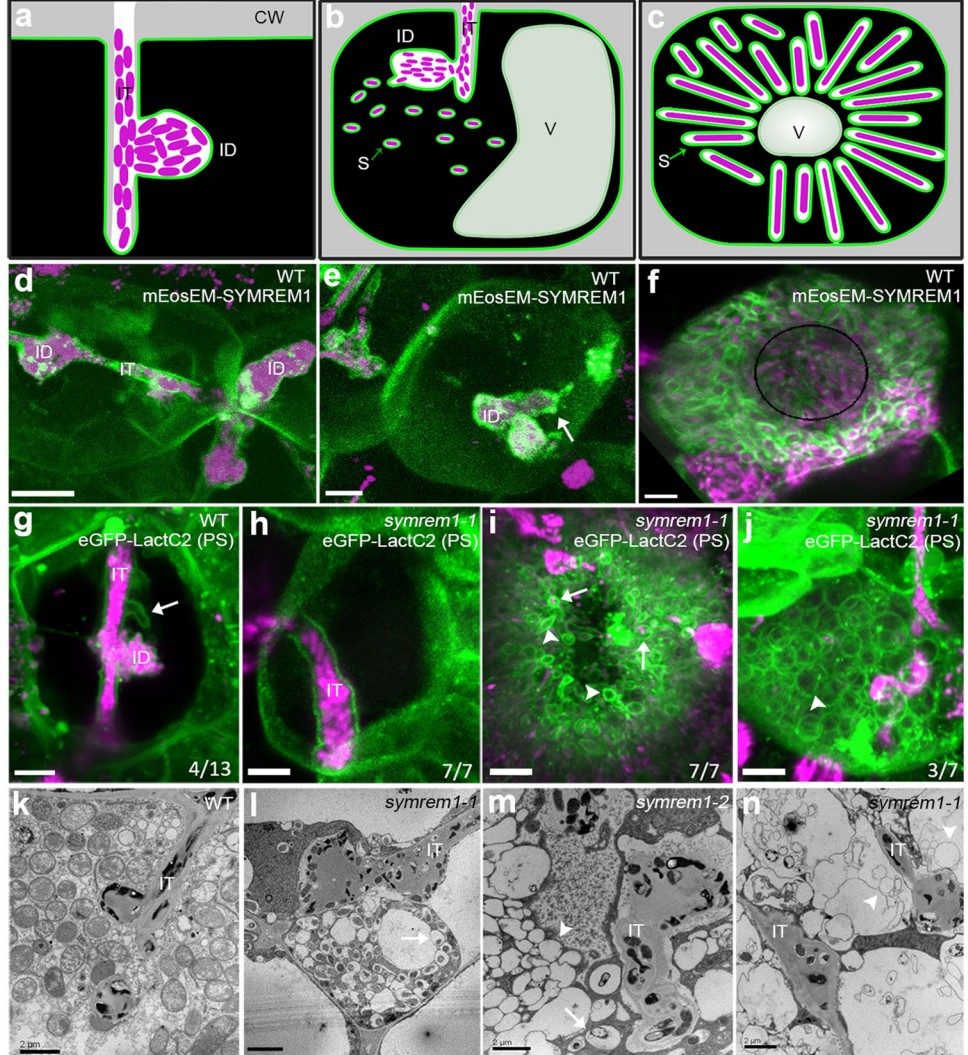

**Fig. 1 | SYMREM1 stabilizes confined membrane tubes during infection.**
**a**–**c** Illustrations indicate an infection thread (IT) with an infection droplet (ID; **a**), symbiosome (S) formation (**b**) and a symbiosome-filled infected cell (**c**); CW cell wall, V vacuole. **d**–**f** Expression of a mEosEM-SYMREM1 fusion protein (green) specifically labels IT membranes (**d**), accumulates at bacterial droplet structures (ID) (**e**), and symbiosome membranes (**f**). *S. meliloti* expressing a mCherry marker is shown in magenta. The white arrow (in **e**) indicates the bacterial release site; the black circle line (in **f**) indicates the central vacuole. Single focal plane images for maximum projections of z-stacks (**d**–**f**) can be found in Supplementary Fig. 1. Experiments were performed with three biological replicates with at least 10 nodules being imaged per replicate. **g**–**j** Membranes were visualized by expressing the phosphatidylserine (PS) biosensor LactC2. Spatially confined membrane tubes (arrow) were found on wild-type (WT) IT containing rhizobia (magenta, **g**) but not on ITs in *symrem1* mutants (**h**). Symbiosome membranes are loosely associated with released rhizobia (arrows, **i**) or appear as empty spheres (arrowheads, **i** and **j**) in *symrem1* mutants. Data were collected based on three biological replicates. Images (**d**–**j**) were taken as z-stacks (internal distance is 0.5 µm) and shown as 3D projections generated by using Imaris. These patterns were confirmed by transmission electron microscopy for WT (**k**) and *symrem1* mutants (**l**–**n**). Scale bars indicate 5 µm (**d**–**j**) and 2 µm (**k**–**n**). IT infection thread, ID infection droplet. The sketches (**a**–**c**) were drawn with Inkscape.

fresh SYMREM1 protein extracts revealed an apparent molecular weight of 57 kDa that matches the calculated size for a dimeric remorin protein (Fig. 2c). Since the filaments were too amorphous for further structural assessment by cryo-EM, we conducted 3D modeling using AlphaFold[44].

In line with previous analyses, AlphaFold predicted a largely disordered N-terminal region encompassing residues 1–69, followed by an α-helical segment from residues 70–187 and a disordered C-terminus at residues 188–205. Interestingly, the search for multimers of SYMREM1 consistently yielded a head-to-tail dimer as a recurrent structural unit (Fig. 2d). Here, the slightly bent α-helical segments of two monomers interacted through extended hydrophobic patches, with the N-termini pointing outwards. Using AlphaFold2[45] predictions for higher-order oligomers we repeatedly obtained alignments in flexible sheets that can be extended to helical structures (Fig. 2e–g).

Our model further indicated a consistent interaction between two dimers that was not mediated by hydrophobic interactions, but rather by a few selected hydrogen bonds. Calculating electrostatic surface potentials on these structures revealed a strong enrichment of positive potentials at one site of the sheet (Fig. 2f). Interestingly, the propagation of these dimeric interactions resulted in the prediction of helical superstructures with flexible inner diameters (Fig. 2g). The disordered N-termini are extending to each side of the helix and are not visualized in the model.

Next, we asked whether the predicted sheet-like and potential super-helical organization of the SYMREM1 oligomer has an impact on membrane topologies along infection threads. For this, we ectopically expressed fluorescently labeled SYMREM1 in WT *M. truncatula* plants, a setup that was previously shown to elevate nodulation levels in *Lotus japonicus*[46]. Here, we observed that the frequency of stabilized

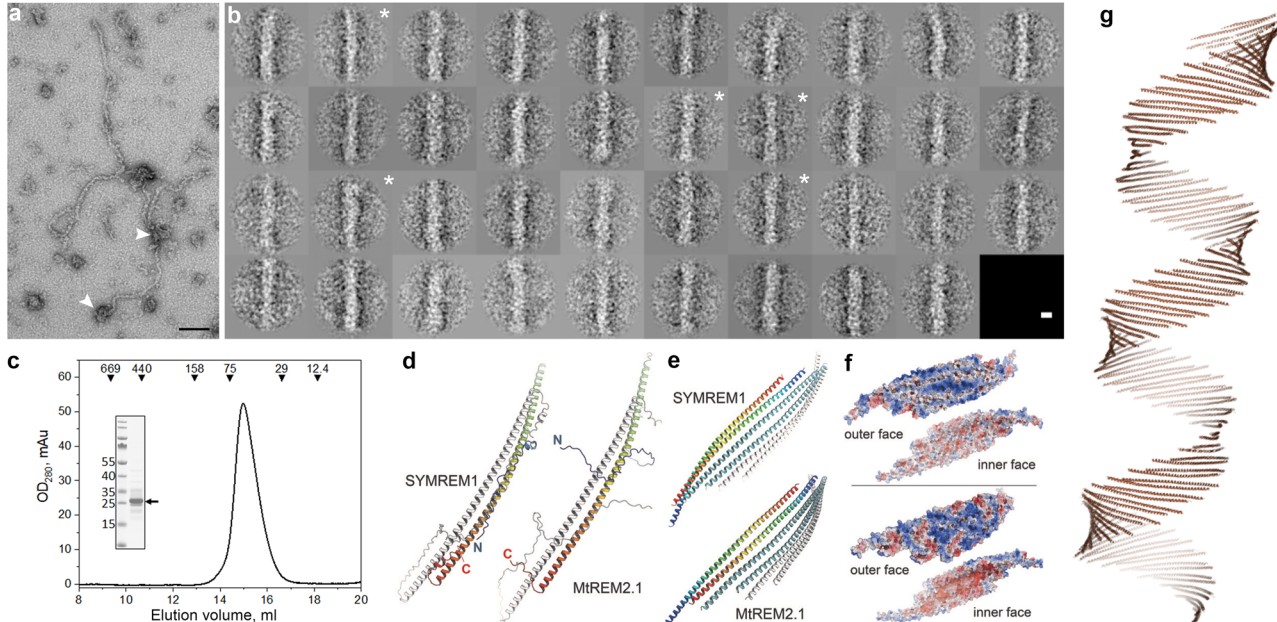

**Fig. 2 | SYMREM1 forms oligomeric alpha helical assemblies. a** Representative raw electron micrograph of purified, recombinant SYMREM1 stained with 2% uranyl acetate. Arrowheads indicate irregular protein bodies. Scale bar indicates 100 nm, experiments were performed twice with independently isolated recombinant SYMREM1 protein. **b** 2D class averages derived from multivariate statistical analysis of all 389 particle images. Each class contains on average ten images. Class averages that show twisted features are marked with a white asterisk. Scale bar indicates 100 Å. **c** Elution profile of recombinant His-SYMREM1. Molecular masses (in kDa) and positions of elution peaks for standard proteins are indicated with triangles on the top. Insert: SDS-PAGE after Coomassie staining of the purified His-SYMREM1 (labeled by an arrow); molecular masses of the pre-stained protein standards are indicated on the left in kDa. **d** AlphaFold predictions for homodimers of SYMREM1

(left) and MtREM2.1 (right). One monomer is colored from blue at the N-terminus to red at the C-terminus, the other in white. The Extended helical regions of both remorins form highly similar, antiparallel dimers. **e** Prediction of higher-order oligomers using AlphaFold2. The remorin homodimers form flexible sheets that can be extended into helical structures. **f** Electrostatic surface potential maps for the two faces of the sheets formed by SYMREM1 (above) and MtREM2.1 (below), contoured from $-5k_BT$ (red) to $+5k_BT$ (blue). In both cases, the convex faces show a positive electrostatic potential, while that of the concave faces is negative. **g** A predicated helical super-structure based on the oligomerization of SYMREM1 dimers. It is important to note that the diameter of this higher order structural prediction is highly variable and depends on the alignment of the individual dimers.

membrane tubes remained unaltered (Fig. 3a) compared to those observed in LactC2-labeled WT cells (Fig. 1g), indicating a temporary nature of these structures at membrane interfaces that continuously release rhizobia. To test this hypothesis, we made use of the release-compromised *ipd3-1* mutant, which is defective in the transcriptional activator CYCLOPS that regulates, among other genes, the expression of endogenous *SYMREM1*[47]. This reported transcriptional regulation also translated into reduced SYMREM1 protein levels in the *ipd3-1* mutant (Supplementary Fig. 4). In line with this and similar to WT plants, PS-labeling revealed membrane tube formation in 36% of all assessed ITs in the *ipd3-1* mutant (Fig. 3b). However, ectopic expression of fluorophore-tagged SYMREM1 in *ipd3-1* significantly increased IT-associated membrane tubulation to 75% with several tubes per IT being frequently observed (Fig. 3c). These data further support a function of SYMREM1 in tubulation of membranes that are not supported by a rigid cell wall.

To address the types of negatively charged lipids that can be bound by the positively charged surfaces of SYMREM1 oligomers, we purified recombinantly expressed and His-tagged SYMREM1 from *E. coli* and hybridized it to a lipid strip containing 15 different spotted lipids. Clear binding of SYMREM1 was observed for phosphatidylinositol (PI), PI(3)P, PI(3,4)P₂, PI(3,5)P₂, and PI(4,5)P₂ while PI monophosphates such as PI(4)P and PI(5)P were not bound by SYMREM1 (Supplementary Fig. 5a). To verify these results independently, we generated giant unilamellar vesicles (GUVs) comprised of PI(4)P, PI(3,5)P₂ or PI(4,5)P₂ and incubated these GUVs with recombinant SYMREM1. While no association was observed with GUVs containing PI(4)P (Supplementary Fig. 5b), SYMREM1 bound to GUVs with PI(3,5) P₂ and PI(4,5)P₂, which resulted in lipid accumulations at these sites

(Supplementary Fig. 5c, d). Although recombinant SYMREM1 associated with these GUVs it should be noted that the protein is not palmitoylated (S-acylated) when being isolated from *E. coli*. This lipidation has, however, been reported to greatly enhance tight membrane binding of SYMREM1[36]. To test the effect of tight membrane association of SYMREM1, we generated GUVs comprised of Ni-NTA lipids (18:1 DGS-NTA(Ni)) that should tightly associate with His-tagged SYMREM1. While His-tagged GFP (control) did not alter the GUVs at all (Fig. 3d), SYMREM1 application resulted in dramatically altered topologies with many membrane blebs being formed (Fig. 3e) that frequently detached as membrane spheres (Supplementary Fig. 6a). We also occasionally observed stabilized membrane invaginations (Supplementary Fig. 6b). We then expanded this analysis and generated recombinant His-tagged SYMREM1 variants comprised of the sole N-terminal intrinsically disordered region (SYMREM1ᴵᴰᴿ, residues 1-73) and the C-terminal region (SYMREM1ᶜᵗᵉʳᵐ; residues 74-205). While the association of SYMREM1ᴵᴰᴿ did not change GUV topologies (Fig. 3f), incubation of GUVs with SYMREM1ᶜᵗᵉʳᵐ had similar effects as observed for full-length SYMREM1 with membrane blebs (Fig. 3g; Supplementary Fig. 6c) and membrane invaginations (Supplementary Fig. 6d) being formed.

To assess whether we can recapitulate these findings in an in vivo system, we expressed these SYMREM1 variants as YFP fusion proteins in cell wall-free *Nicotiana benthamiana* mesophyll protoplasts. Such cells adopt a spherical shape under mild hypotonic conditions as visualized by expressing a generic membrane marker probe (mCitrine-LTI6b; Supplementary Fig. 7a). However, when expressing full-length SYMREM1, we observed numerous tubular outgrowths developing on 64 out of 112 (57%) inspected protoplasts (Fig. 4a–c, Supplementary Fig. 7b). The formed tubular structures greatly varied in length with the

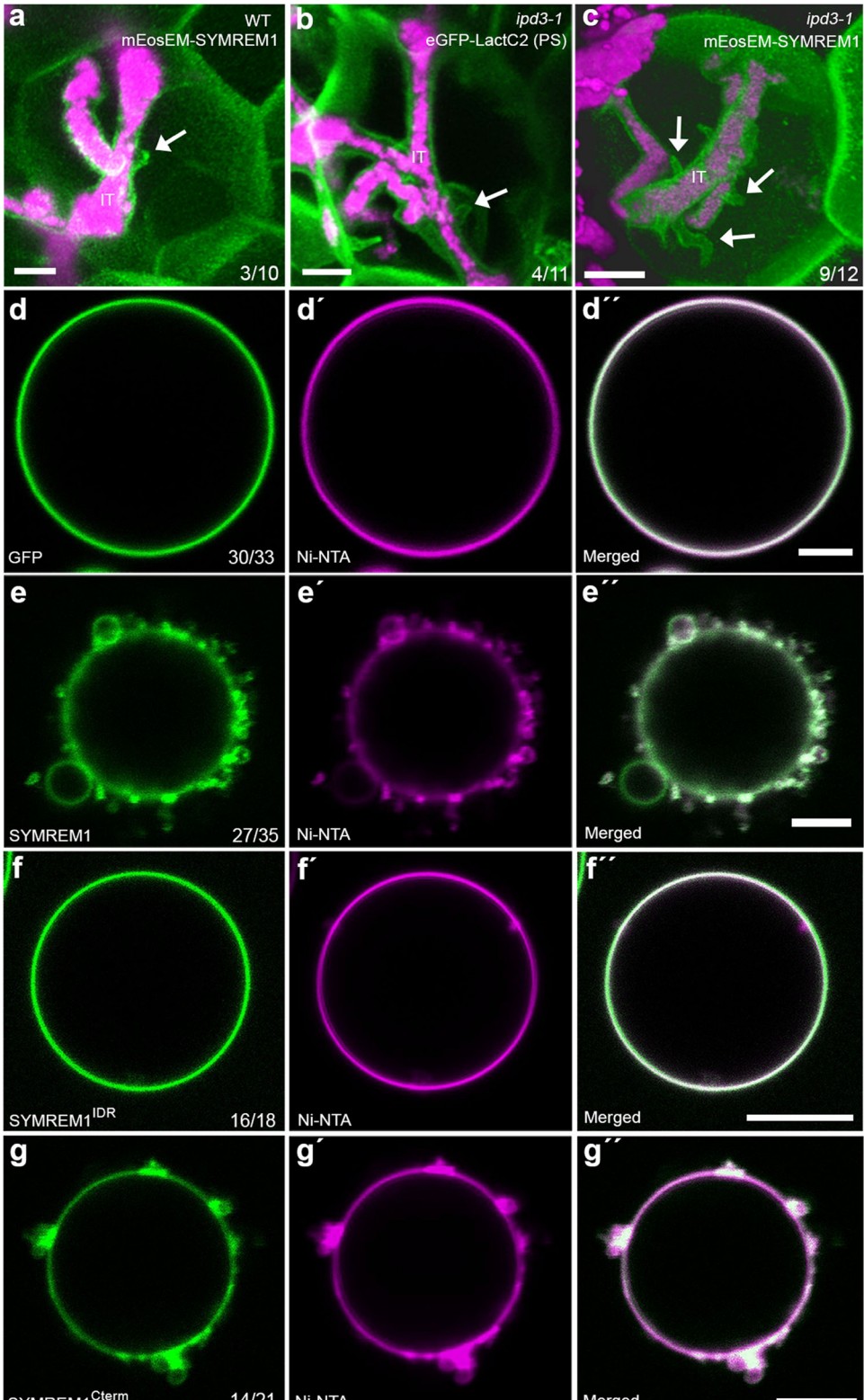

**Fig. 3 | SYMREM1 can induce membrane morphology changes. a–c** Membrane tubes were found on nodular ITs in the *ipd3-1* mutant (arrow, **b**), while ectopic expression of SYMREM1 greatly increased these tubular outgrowths in the *ipd3-1* mutant (arrows, **c**) but not on WT (arrow, **a**). Membranes were visualized by expressing mEosEM-SYMREM1 (**a, c**) or the phosphatidylserine (PS) biosensor LactC2 (**b**), both shown in green. IT infection thread. *S. meliloti* expressing a mCherry marker is shown in magenta. Data were collected based on three biological replicates. All images were taken as z-stacks (internal distance is 0.5 μm) and are shown as 3D projection generated by using Imaris. **d–g** Application of His-tagged GFP (control, green) did not alter the morphology of Ni-NTA containing GUVs (Atto 647N-DOPE was used as membrane marker; magenta) (**d–d"**). By contrast, application of His-GFP tagged SYMREM1 (green, **e-e"**) and His-GFP tagged SYMREM1Cterm (green, **g–g"**) resulted in dramatically altered topologies with many membrane blebs being formed. This was not observed when applying the His-GFP-tagged SYMREM1IDR variant (green, **f–f"**). GUV experiments were performed twice and showed similar results. Scale bars indicate 5 μm (**a–c**) and 10 μm (**d–g**).

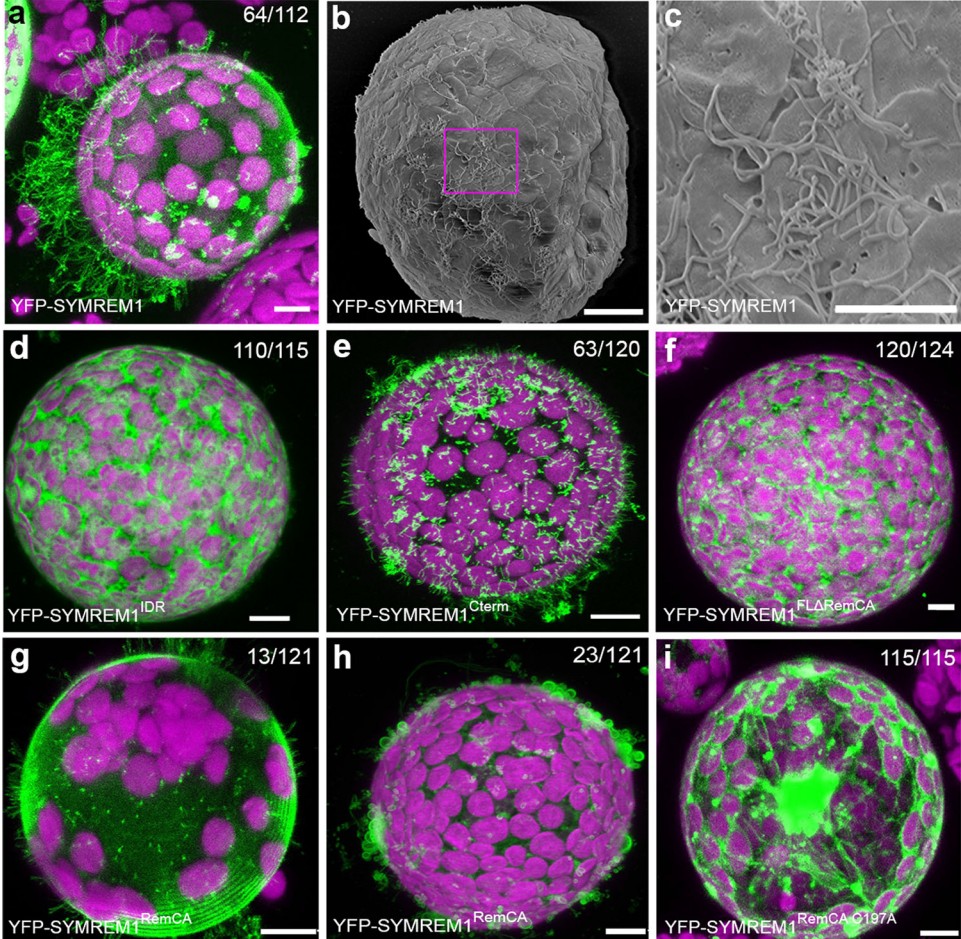

**Fig. 4 | SYMREM1 stabilizes membrane tubulation and curvature in a cell wall-independent manner. a–c** *N. benthamiana* leaf protoplasts ectopically expressing YFP-SYMREM1 develop multiple membrane tubes as shown by confocal laser-scanning microscopy (**a**) and scanning electron microcopy (**b**) and close-up in (**c**). **d–i** Protoplasts expressing different SYMREM1 variants: only IDR domain (SYMREM1[IDR], **d**), only C-terminal including RemCA (SYMREM1[Cterm], **e**), full-length sequence lacking the RemCA domain (SYMREM1[FLΔRemCA], **f**), RemCA domain only (SYMREM1[RemCA], **g**, **h**), and the RemCA domain bearing a mutation of the palmitoylated Cys197 residue (SYMREM1[RemCA C197A], **i**). Scale bars indicate 10 µm. All the confocal images are shown as maximal projections. Magenta signals derive from chlorophyll autofluorescence within chloroplasts. Numbers indicate frequencies of all observations. Protoplast isolation experiments were at least performed in four biological replicates, with similar tube formation frequencies found in each replicate.

longest ones being more than 70 mm and occasionally branched (Supplementary Fig. 7c). Scanning electron microscopy (SEM) (Fig. 4b, c) revealed an average diameter of these protrusions of $0.18 \pm 0.03$ mm (Supplementary Fig. 7d). Expression of SYMREM1[IDR] in protoplasts failed to induce and stabilize these tubes as verified by confocal laser-scanning microscopy (Fig. 4d) and SEM (Supplementary Fig. 7e, f). By contrast, expression of the helical C-terminal region (SYMREM1[Cterm]) phenocopied patterns observed for the full-length protein (Fig. 4e) even though the tubes were mostly shorter. Taken together, these observations are consistent with those from GUVs.

Since SYMREM1 associates with the plasma membrane via the amphipathic and often palmitoylated RemCA peptide (amino acids 171–205)[36], we tested the impact of these C-terminal 35 amino acids on membrane topologies. Truncating RemCA from full-length SYMREM1 (SYMREM1[FLΔRemCA], residues 1–170) resulted in a cytosolic protein that was unable to induce and/or stabilize these membrane outgrowths (Fig. 4f). Expression of the RemCA membrane anchor alone (SYMREM1[RemCA]) was sufficient to drive membrane topology changes. While 13 out of 121 protoplasts developed tubes on the protoplast surface (Fig. 4g), 23 out of 121 protoplasts exhibited less confined and more bulky membrane blebs (Fig. 4h). Mutating the palmitoylated Cys197 residue in RemCA (SYMREM1[RemCA C197A]) fully abolished membrane association of the peptide and its impact on membrane topology

(Fig. 4i). These data show that the RemCA peptide alone is sufficient for initiating membrane tubulation but cannot maintain and/or stabilize long protrusions at high-frequency.

## SYMREM1 indirectly associates with actin

As tubular membrane outgrowths such as pollen tubes and root hairs are usually actin-dependent, we co-expressed full-length SYMREM1 with the actin marker LifeAct in protoplasts and observed that all tubes contained central actin filaments (Fig. 5a). The importance of this was additionally supported by the fact that application of cytochalasin D, an actin depolymerizing agent, dramatically decreased the number of tubular outgrowths (11 out of 118 protoplasts) in protoplasts expressing SYMREM1 (Fig. 5b). Instead, upon actin depolymerization we observed the same kind of membrane blebs (Fig. 5c) as induced upon expression of the SYMREM1[RemCA] peptide (Fig. 4h). By contrast, the application of the microtubule de-polymerizing agent oryzalin did not alter membrane tube formation (Fig. 5d). Furthermore, expression of the symbiotic and actin-associated formin protein SYFO1[14] resulted in a tip-localized signal at these protrusions (Fig. 5e) with SYFO1 being recruited to SYMREM1 foci over time (Supplementary Fig. 8a). Interestingly, SYFO1 was also present in membrane blebs induced by either expressing SYMREM1 and subsequent cytochalasin D treatment (Fig. 5f) or when solely expressing

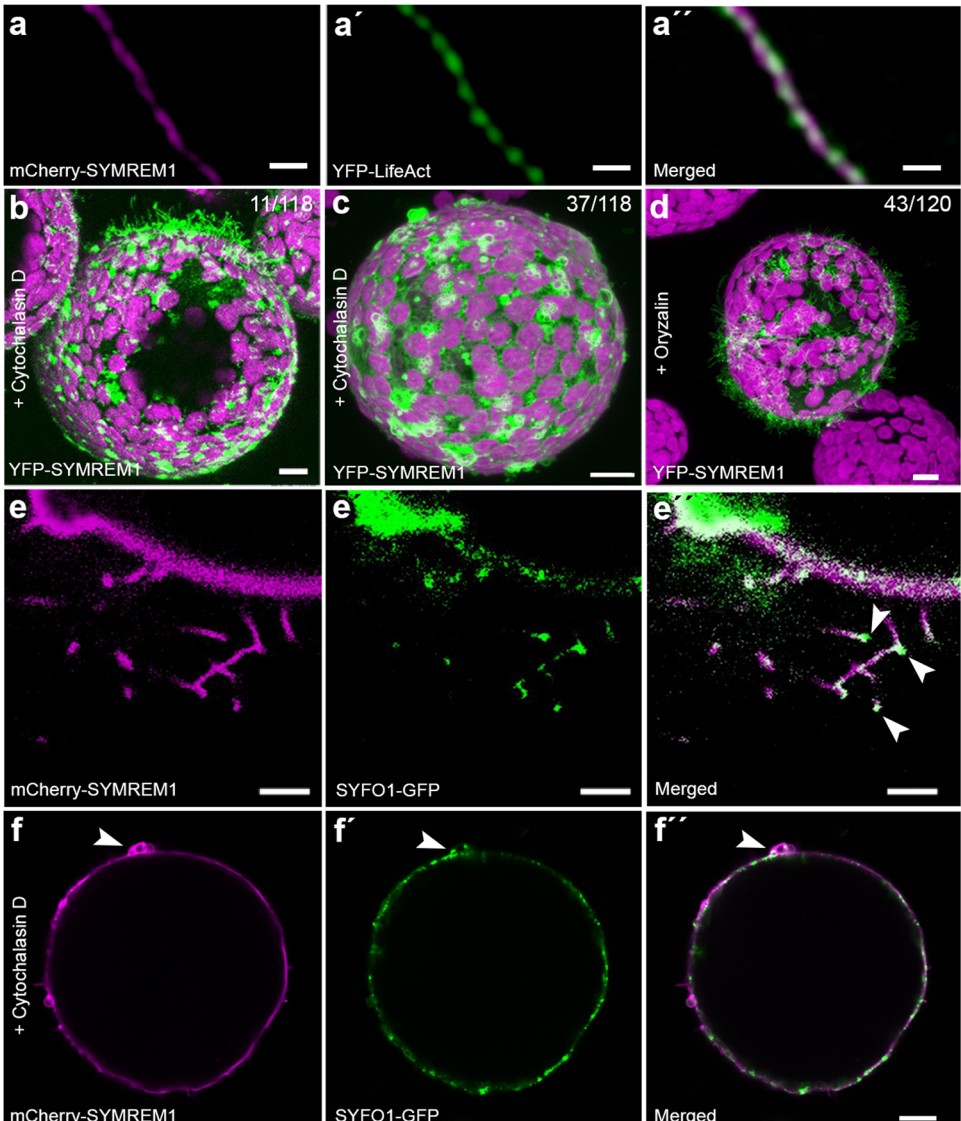

**Fig. 5 | The formation of membrane tubes is actin-dependent. a–a′′** *N. benthamiana* leaf protoplasts ectopically expressing mCherry-SYMREM1 (magenta) developed multiple membrane tubes that comprised a central actin strand as labeled by LifeAct (green). **b–d** Protoplasts ectopically expressing YFP-SYMREM1 and treated with Cytochalasin D (**b, c**) and Oryzalin treatment (**d**). Magenta signals in (**b–d**) derived from chlorophyll autofluorescence within chloroplasts. Numbers indicate frequencies of observations and images are shown as maximal projections.

**e–f′** *N. benthamiana* protoplasts ectopically expressing mCherry-SYMREM1 (magenta) developed multiple membrane tubes with a tip localized formin protein SYFO1 (green, white arrowheads, **e–e′′**). SYFO1 remained to be recruited into membrane blebs after Cytochalasin D treatment of these protoplasts (green, white arrowheads, **f–f′**). Scale bars indicate 10 μm. Data were obtained from four independent biological replicates.

the RemCA peptide (Supplementary Fig. 8b) while no actin strands were found in these blebs (Supplementary Fig. 8c). Since cytochalasin D treatment of SYMREM1-expressing protoplasts severely inhibited membrane tubulation we concluded that SYMREM1 stabilizes rather than actively drives outgrowth of these membrane tubes in an actin-dependent manner.

To investigate whether SYMREM1 and actin directly interact, we performed co-sedimentation assays. To demonstrate functionality of the setup, we first incubated actin with the known actin-binding formin protein mDia1-ct from human cells[48]. In the control samples G-actin and mDia1-ct were predominantly present in the pellet and the supernatant, respectively, while incubation of both proteins resulted in an enrichment of mDia1-ct in the pellet, indicative of actin-binding (Supplementary Fig. 9a). This phenomenon was neither observed when mixing 10 or 30 μM SYMREM1 with actin since SYMREM1 remained in the soluble fraction (Supplementary Fig. 9b, c). These data

suggest that actin and SYMREM1, at least in vitro, do not interact directly with each other.

To test the association of SYMREM1 with actin under symbiotic conditions, we conducted co-labeling experiments on nodule sections, where we labeled actin with either Phalloidin or by using stable transgenic plants expressing an actin reporter (ABD-mCherry). SYMREM1 was labeled by a specific antibody[35] or by expressing a fluorescently tagged SYMREM1 protein. In these experiments, SYMREM1 was found along infection threads within the nodular fixation zone (Fig. 6a, Supplementary Fig. 10) with actin being ubiquitously present in all cells. To gain a more detailed view on this, we applied structured illumination super-resolution microscopy (SIM) to these samples. Actin and SYMREM1 tightly aligned but were slightly displaced to each other along nodular ITs (Fig. 6b–d). A similar situation was found in the nodular fixation zone, where both proteins showed the same association pattern (Fig. 6e–h; Supplementary Figs. 10, 11).

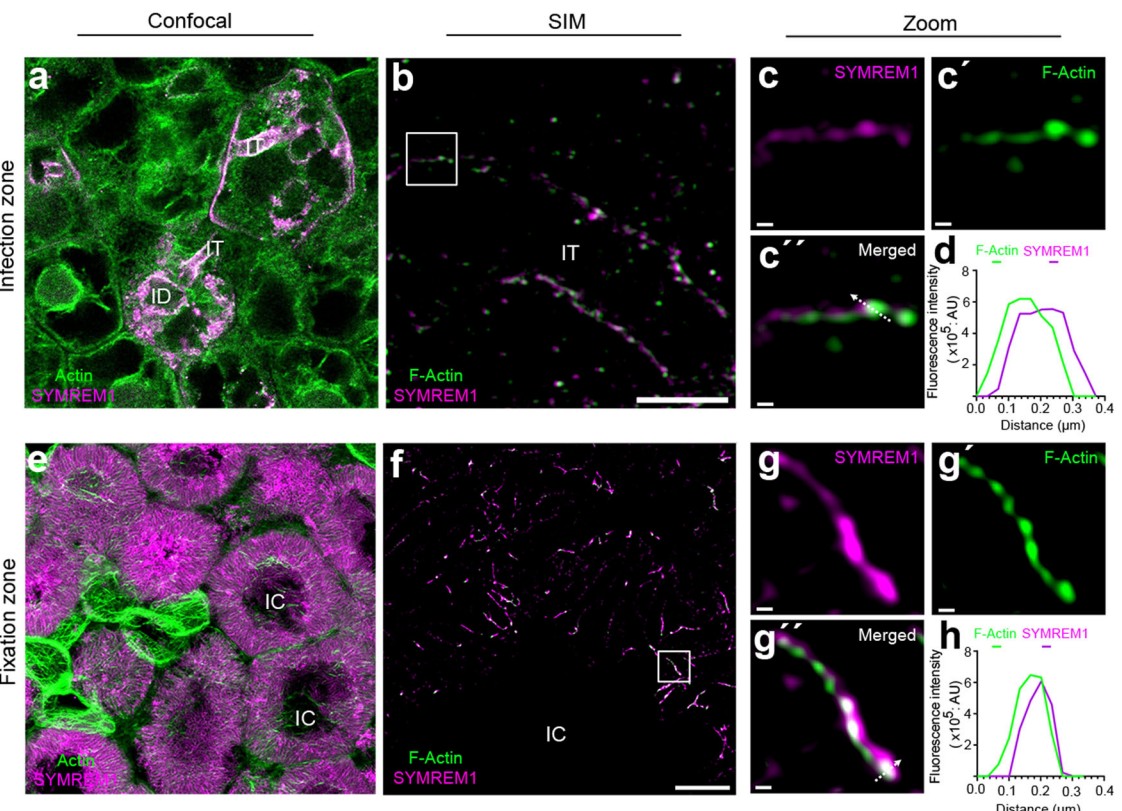

**Fig. 6 | SYMREM1 and F-actin colocalize at infection structures inside nodules.**
Sections of two weeks old nodules from a stable transgenic *Medicago truncatula* plant expressing an ABD-mCherry actin reporter (green). **a**, **e** SYMREM1 was immunolabelled (magenta) using a specific α-SYMREM1 antibody. Maximum intensity projection of an infection thread (IT) (**b**) and symbiosomes (**f**) immunolabelled for SYMREM1 (magenta), counterstained with Phalloidin (green) and imaged using super-resolution structured illumination microscopy (SIM). **c**–**c″**, **g**–**g″** Close-ups of areas indicated by white boxes in (**b**, **f**). **d**, **h** 2D linescans of the transect indicated by the white arrow in (**c″**, **g″**). Images are maximum projections (**a**, **b**, **e**) or single plane SIM images (**f**, **c**–**c‴**, **g**–**g‴**). Structures depicted in (**f**–**g″**) derived from symbiosome membranes for which the raw images are provided in Supplementary Fig. 11. Scale bars indicate 5 μm (**a**, **b**, **f**) and 200 nm in (**c**, **g**). For confocal analysis (**a**, **e**), data were collected based on three biological replicates, with at least eight nodules being assessed for each replicate. For SIM analysis, two biological replicates were performed and three nodules were assessed for each replica.

## Membrane scaffolding is conserved among remorin proteins

To further test whether SYMREM1 is able to stabilize exogenously induced membrane deformations and functions as a stabilizing scaffold exclusively for positive curvatures, we assessed whether negative curvatures, as underlying IT-associated membrane tubes, were also maintained in the presence of this protein. Therefore, we isolated SYMREM1-expressing protoplasts and immediately indented them with a micro-capillary for 30 min. Here, 10/13 protoplasts expressing the LTI6b membrane marker (control) re-inflated directly after releasing the pressure (Supplementary Fig. 12a). This indicates that the cytoskeleton alone is not sufficient to dynamically maintain these short-term indentations as shown for protoplasts that had been confined in controlled geometries[49]. By contrast, full re-inflation was observed in only 4/14 protoplasts expressing SYMREM1, while microcapillary-induced membrane deformations were maintained in the majority (10/14) of these protoplasts (Supplementary Fig. 12b). Such stabilization was also observed in most protoplasts expressing the truncated SYMREM1^Cterm variant (Supplementary Fig. 12c) while the RemCA peptide alone (SYMREM1^RemCA) was not sufficient to mediate this phenomenon (Supplementary Fig. 12d). These data suggest that topology scaffolding of SYMREM1 is highly flexible and does not rely on an invariable protein structure.

As a last set of experiments, we assessed whether membrane topology scaffolding is limited to SYMREM1 or a more wide-spread feature within the remorin protein family. In analogy to the Medicago SYMREM1 protein, expression of the orthologous gene from *Lotus japonicus* (*LjSYMREM1*; Lj4g3v2928720) resulted in stabilized membrane tubes at high frequency (Supplementary Fig. 13a). Evolutionary, the presence of SYMREM1 proteins is mainly restricted to species that maintained the root nodule symbiosis and species that do not support intracellular infection of symbionts, either fungal or bacterial, through their epidermis have lost *SYMREM 1* (Supplementary Fig. 14). In addition, the only non-symbiotic clade that retained *SYMREM1*, the Caryophyllales, experienced bursts of diversifying positive selection indicative of neofunctionalization (Supplementary Data 1), suggesting that the purifying selection maintaining *SYMREM1* in plant genomes is linked to the maintenance of symbiotic abilities. Interestingly, a closely related protein (MedtruChr5g0397781; MtREM2.1) shows an expanded phylogenic distribution, while both *SYMREM1* and *MtREM2.1* derive from a Papilionoideae duplication. To assess whether MtREM2.1 has similar impact on membrane topology as shown for SYMREM1, we expressed a MtREM2.1 fusion protein in mesophyll protoplasts. Indeed, the presence of this protein resulted in the stabilization of numerous membrane tubes (Supplementary Fig. 13b), and this function is supported by AlphaFold predictions that revealed a putative structure that is highly similar to the one predicted for SYMREM1 (Fig. 2d–f). To address whether MtREM2.1 localizes to highly curved membranes, we first surveyed its gene expression profile by using global transcriptome data, which revealed an induced expression during endosymbiotic interactions[50]. We independently confirmed this by quantitative Real-Time PCR on cDNA generated from nodulated and mycorrhized roots (Supplementary Fig. 15a). To increase the

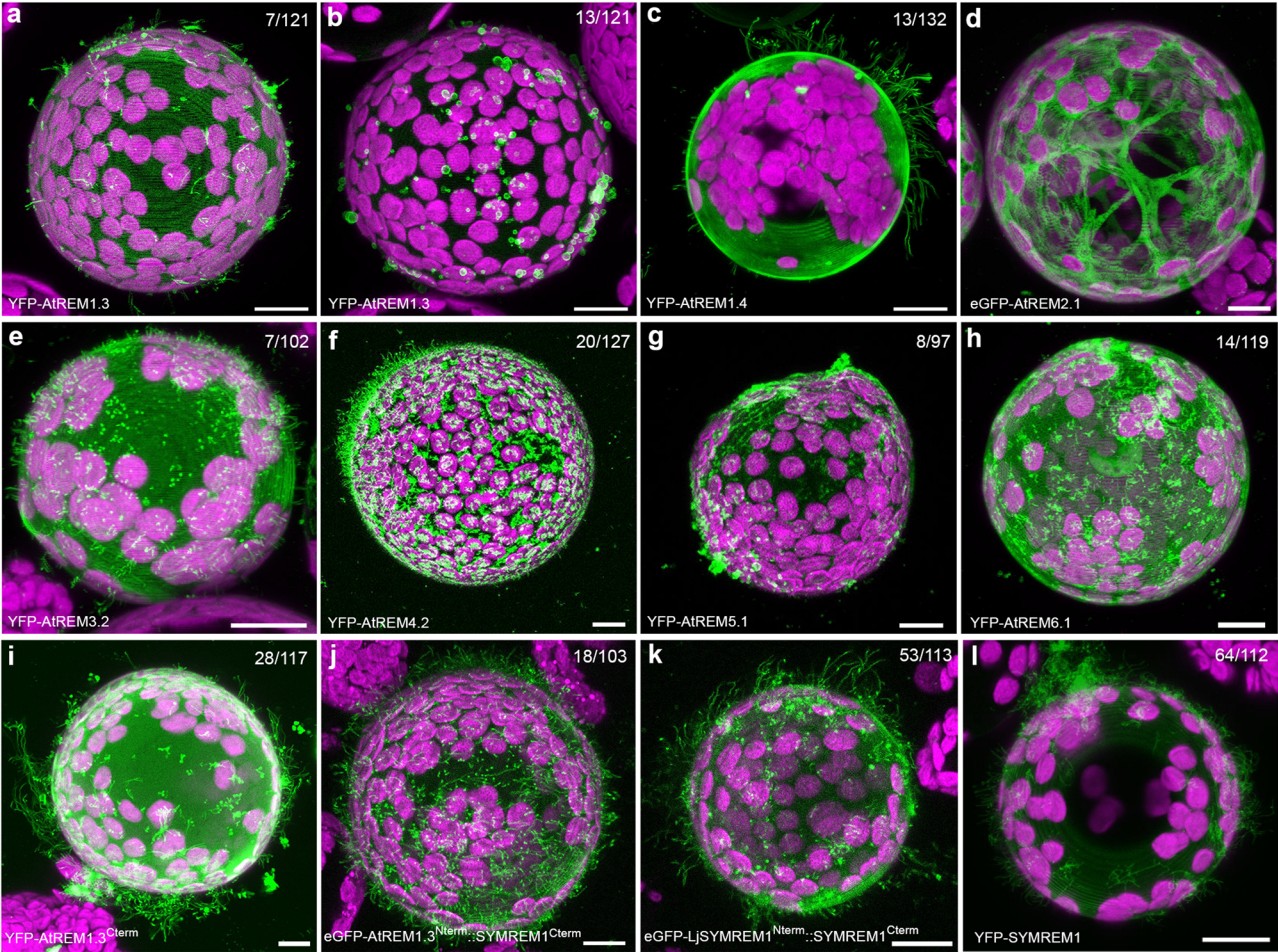

**Fig. 7 | Remorin-induced membrane tubulation in protoplasts. a–h** *N. benthamiana* leaf protoplasts ectopically expressing different remorins from the non-symbiotic plant *Arabidopsis thaliana* covering all sub-groups of the Arabidopsis remorin family. AtREM1.3 (**a**, **b**), AtREM1.4 (**c**), AtREM2.1 (**d**), AtREM3.2 (**e**), AtREM4.2 (**f**), AtREM5.1 (**g**), AtREM6.1 (**h**). **i–k** Membrane tubulation activity was also assessed for a N-terminally truncated variant of AtREM1.3 (AtREM1.3$^{Cterm}$, **i**), a chimeric protein comprised of the AtREM1.3 N-terminal and the SYMREM1

C-terminal region (AtREM1.3$^{Nterm}$::SYMREM1$^{Cterm}$, **j**), a dual-legume SYMREM1 chimera (LjSYMREM1$^{Nterm}$::SYMREM1$^{Cterm}$, **k**) and SYMREM1 (**l**). Scale bars indicate 10 μm. All images are shown as maximal projections. The magenta signal indicates chlorophyll autofluorescence from chloroplasts. Numbers indicate the frequencies of the presented observations across at least three independent biological replicates.

spatio-temporal resolution of this analysis, we generated a transcriptional reporter with a 1 kb long fragment of the putative *MtREM2.1* promoter driving the expression of a β-glucuronidase gene (GUS). *MtREM2.1* promoter activity was detected in young and maintained in the apical region of mature nodules (Supplementary Fig. 15b–d). In addition, we found increased GUS activity in cells containing arbuscules when inoculating roots with the arbuscular mycorrhizal fungus *Rhizophagus irregularis* (Supplementary Fig. 15e, f). To assess the localization of the MtREM2.1 protein, we generated *Medicago truncatula* stable lines (WT R108 background) expressing an mCherry-MtREM2.1 fusion protein under the control of the endogenous promoter. In roots inoculated with *R. irregularis*, MtREM2.1 predominantly localized to the host-derived and highly curved membrane (peri-arbuscular membrane) surrounding intracellular hyphae, from their early penetration into inner cortical cells (Supplementary Fig. 15g) until the development of fully branched arbuscules (Supplementary Fig. 15h). Interestingly this arbuscular membrane is, similarly to infection droplets or symbiosome membranes, not supported by a rigid cell wall[51].

While all three tested group 2 remorins (Medicago SYMREM1, LjSYMREM1, and MtREM2.1) stabilized membrane tubes on protoplasts, systematic expression of different members of the

Arabidopsis remorin family, confirmed this ability for AtREM1.3, AtREM1.4, and AtREM4.2 even though membrane tubulation was stabilized with much lower frequencies in all cases (Fig. 7a–c, f). In addition, and similar to the effect observed in the presence of RemCA (Fig. 4h) or cytochalasin D (Fig. 5c), expression of AtREM1.3 resulted in membrane bleb formation in 11% of the cases. No such effects were observed for a so far uncharacterized remorin (AtREM2.1; AT1G63295), which was mostly cytosolic (Fig. 7d), whereas expression of AtREM3.2, AtREM5.1, and AtREM6.1 led to focal accumulations and the formation of shorter membrane tubes (Fig. 7e, g, h). These results indicate that the ability of remorins to act as membrane topology scaffolds is conserved in land plants. Besides this, it was intriguing to note that expression of an N-terminally truncated variant of AtREM1.3 lacking its IDR significantly increased the ability to stabilize membrane tubes from 6% to 24% (Fig. 7i), suggesting an inhibitory role of the disordered N-terminal region. This hypothesis was supported by expressing a chimeric protein comprised of the AtREM1.3 N-terminal and the Medicago SYMREM1 C-terminal region (AtREM1.3$^{N-term}$::SYMREM1$^{C-term}$) that reduced membrane tubulation frequencies from 57% (Figs. 4a, 7l) to 17% (Fig. 7j), while expression of a LjSYMREM1/SYMREM1 chimera maintained high tubulation levels at 47% (Fig. 7k).

Taken together, these results show that the largely legume-confined remorins SYMREM1 and MtREM2.1 represent evolutionary optimizations to structurally support curvatures of cell wall-devoid membranes throughout endosymbiotic infection processes.

## Discussion

Plant membranes are usually supported by a rigid cell wall that counteracts the intracellular turgor pressure and maintains, together with cortical microtubules and actin, cellular shapes over their lifespan[49,52–54]. As all cell wall constituents are either secreted to the apoplast or synthesized directly onto the extracellular leaflet of the plasma membrane, these responses require time. Furthermore, subsequent loosening of the rigidified cell wall to adopt temporal changes in membrane topologies mostly account for protrusion-like shapes. In contrast, larger scale and negatively curved membrane invaginations as found during microbial infections of host cells may require other types of scaffolding prior to de novo cell wall apposition. Such functions can be maintained by scaffold proteins as exemplified during clathrin-mediated endocytosis, where the adaptor complex AP2 together with clathrin light and heavy chains stabilizes endocytotic vesicles[22,23]. In addition, multimeric and helical forming ESCRT-III subunits predominantly contribute to membrane remodeling and cytokinetic abscission[55,56]. In plants, further evidence has been presented that endocytosis of membrane nanodomain-localized proteins might occur via a flotillin- and remorin-dependent but clathrin-independent pathway[57–59]. However, compelling evidence for topological scaffolding functions of these proteins has been missing, while the impact of remorins on membrane fluidity is experimentally supported[28].

Here, we demonstrate that plant remorin proteins can stabilize membrane topologies independent of a cell wall with legume group 2 remorins, which are mostly found in plants that have maintained a symbiotic association with (both) rhizobia and arbuscular mycorrhiza fungi, showing the strongest effect on membrane tubulation (Fig. 4; Supplementary Fig. 13). This makes sense since both endosymbioses require large-scale membrane invaginations during intracellular colonization and stabilization of the symbiotic membranes. Here, SYMREM1 is required for stabilizing the IT and the symbiosome membrane (Fig. 1d–f; Supplementary Fig. 1g–i). By contrast, MtREM2.1 is potentially involved in stabilizing the periarbuscular membrane enclosing mycorrhizal hyphae (Supplementary Fig. 15g, h). Although cell wall components are deposited within the older parts of ITs[20], the initial curvature needs to be differently stabilized and additionally maintained around cell wall-free bacterial release sites. In contrast to membrane bending proteins belonging to the Bin/Amphiphysin/RVS (BAR) domain family that mediate either positive of negative curvature[60] and that are, amongst others, involved in autophagosome formation[61] and cell plate formation[62] in plants, SYMREM1 seems capable of stabilizing both types of directions (Fig. 4; Supplementary Figs. 6; 12). This feature is mainly mediated by the evolutionary conserved C-terminal region of the protein, which is additionally required for protein oligomerization into dimers via hydrogen bonding and hydrophobic interactions (Fig. 2d). Membrane association is further assisted by a palmitoylation of a C-terminal cysteine residue[36] (Fig. 4i). We propose that the required forces are provided by a high membrane binding energy and internal rigidity of the SYMREM1 scaffold that exceed the energy required for membrane bending and membrane rigidity, respectively, as suggested for other scaffold proteins[63]. This assumption is supported by our AlphaFold model predicting curved dimers and multimers (Fig. 2d–g) and by the fact that accumulated shallow insertions of membrane scaffolds, as also mediated by the RemCA anchor[27] and palmitoylation for SYMREM1, have been generally shown to be effective to induce membrane curvatures[64]. Furthermore, the formation of higher order protein sheets or filaments has been shown to occur in vitro[31] (Fig. 2a, b). The impact of such

protein lattices on membrane curvature has also been demonstrated for the human N-Bar protein Endophilin that can adopt higher-order arrangements[65]. Similar observations have also been reported for the F-Bar protein Imp2 from fission yeast, where a helical alignment of Imp2-subunits results in membrane tubulation of human Cos-7 cells[66]. Whether SYMREM1 can form such helical superstructures as predicted (Fig. 2g) and shown for Imp2 and ESCRT-III proteins[55,56] remains to be investigated. Potential hints have recently been provided for the group 1 remorin AtREM1.3 when being expressed in human Cos-7 cells where AtREM1.3 was found to helically wrap around actin strands[67]. Evidence for such tight association of actin and SYMREM1 is also provided in this study, where we showed that membrane tube elongation in protoplasts is dependent on an intact actin cytoskeleton (Fig. 5) and likely driven by formin-mediated actin polymerization as also shown for filopodia growth and retraction[68]. During innate immune responses in Arabidopsis the control of formin condensation is also mediated by remorins[38]. Furthermore, SYMREM1 and actin tightly align alongside symbiotic membranes (Fig. 6; Supplementary Fig. 10). However, a direct interaction between SYMREM1 and filamentous actin could not be demonstrated in vitro (Supplementary Fig. 9), whereas such interactions have been shown for the group 6 remorin GSD1 from rice[69].

In summary, we hypothesize that SYMREM1 provides a scaffolding structure that enable the stabilization of large-scale membrane topologies as found during bacterial release at nodular infection threads (Fig. 1g). We favor a model in which shallow insertions of oligomeric and rigid remorin scaffolds provide the forces for membrane bending and stabilization that may, however, be assisted but other, yet unknown, proteins. In our hands, a membrane-bending model exclusively based on protein crowding is less likely. While it was reported that sole crowding of any membrane-associated protein when exceeding a membrane coverage of more than 20% induces membrane tubulation in GUVs[70], this was not confirmed in our study as neither membrane-immobilized GFP nor SYMREM1$^{IDR}$ (Fig. 3d, f) or insertion of the transmembrane protein LTI6b into protoplasts (Supplementary Fig. 7a) induced these effects. In vivo, such tubulations and invaginations, are sufficient to drive cytoskeleton polarization and consequently targeted secretion of proteins to these sites. In symrem1 mutant nodules, the lack of such polarization may therefore prevent the secretion of proteins required for bacterial release.

Taken together we unraveled a novel mechanism that allows plant cells to stabilize distinct membrane topologies in the absence of a cell wall by an interplay between oligomeric remorin scaffolds and the cytoskeleton. Consequently, remorins may serve roles additional to those of BAR-domain, ESCRT or other scaffold proteins[71].

## Methods

### Plant growth and Rhizobia inoculation

Seeds of *M. truncatula* were surface sterilized by covering them with pure sulfuric acid ($H_2SO_4$) for 10 min, followed by 4–6 times washing with sterile water. Seeds were then covered with bleaching solution (12% NaOCl, 0.1% SDS) for no longer than 60 seconds and washed 4 to 6 times with sterile water. After surface sterilization, seeds were transferred to 1% agar plates and stratified at 4 °C for 3 days in darkness. Germination was allowed for up to 24 h at 24 °C in darkness. The seed coat was removed and seedlings were transferred to plates containing Fahräeus medium supplemented with 0.5 mM $NH_4NO_3$. One week later, they were transferred onto fresh Fahräeus medium without nitrate, but containing AVG (0.1 μM). Inoculations were performed after 4 days of growth on plates without nitrate. For inoculation of *M. truncatula* roots, a *S. meliloti* (Sm2011) liquid culture was centrifuged (3 min, 3000 rpm), washed once with liquid Fahräeus medium and resuspended in liquid Fahräeus medium to a final $OD_{600} = 0.03$. Each root was covered with 1 ml of rhizobia suspension, which was removed after 6 minutes. Afterwards, the plants were placed in a controlled environment chamber at 24 °C with a 16/8 h light/dark photoperiod,

keeping the roots in the dark, for 3 weeks before harvesting the nodules.

## Hairy root transformation and rhizobial inoculation

*M. truncatula* hairy root transformation was performed as previously described[72]. Briefly, transgenic *Agrobacterium rhizogenes* (ARqua I), carrying the plasmid of interest, was grown in LB liquid culture for one day and 300 µl of the culture were spread on LB agar plates supplemented with the corresponding antibiotics for selection, and incubated on plates for two more days before transformation. *M. truncatula* seeds were prepared as mentioned above. After germination the seed coat was removed from the cotyledons of the seedlings under water, and the root meristem was cut off with a scalpel. Cut seedlings were dipped on the Agrobacterium plates and transferred onto solid Fahräeus medium (containing 0.5 mM $NH_4NO_3$). Transformed seedlings were incubated for three days at 22 °C in darkness, following 4 days at 22 °C in white light but keeping the roots in the dark. One week after transformation, seedlings were transferred onto new Fahräeus medium (0.5 mM $NH_4NO_3$) and grown for another 10 days at 24 °C in a controlled environment chamber with 16 h/8 h light/dark photoperiod. Afterwards, the roots were screened to examine the transformation efficiency using a stereomicroscope to detect the corresponding fluorescent signal. Untransformed roots were cut off and plants showing fluorescence roots were transferred to pots (2 plants per pot) having a mixture of equal volume of quartz-sand and vermiculite. All pots were individually watered with liquid Fahräeus medium (without nitrate) and tap water once a week. After 3–5 days, the pots were inoculated with *S. meliloti* ($OD_{600}$ = 0.003).

## Evolutionary analysis

Homologs of MtSYMREM1 (Medtr8g097320.2 or MtrunA17Chr8g0386521) were retrieved using the tBLASTn v2.11.0+[73] against a database of 189 species covering all Viridiplantae lineages with default parameters and e-value threshold of 1e−10. Coding sequences of putative homologs were aligned using MAFFT v7.471[74] with default parameters. Positions with more than 80% of gaps were removed from the subsequent alignment using trimAl v1.4rev15[75] and cleaned alignment subjected to phylogenetic analysis using Maximum Likelihood approach. Prior tree reconstruction, best-fitting evolutionary model was tested using ModelFinder[76] according to the Bayesian Information Criteria. Maximum likelihood analysis was conducted using IQ-TREE2 v2.0.3[77]. Branches support were estimated using 10,000 replicates of both sh-aLRT[78] and UltraFast Bootstraps[79]. Analysis of the main tree revealed that MtSYMREM1 clade derives from Eudicots duplication and then, a subtree corresponding to the Eudicots clade of MtSYMREM1 has been reconstructed based on protein sequences aligned using MUSCLE v3.8.1551[80] and phylogenetic procedure described above. Trees were visualized using the iTOL v6 platform[81].

To look for specific selective pressure acting on the Caryophyllales clade, which is composed only of species not forming infection threads, we conducted branch and branch site analysis. We estimated relaxation ($K < 1$) or intensification ($K > 1$) parameters and also positive selection acting on Caryphyllales (Supplementary Data 1), we used the RELAX and aBSREL programs implemented in the HYHPY software[82,83]. These methods calculate different synonymous and nonsynonymous substitution rates ($\omega = dN/dS$) using the phylogenetic tree topology for both foreground and background branches. Translated CDS of MtSYMREM1 homologs were aligned using DECIPHER[84]. In total, 191 sequences and 162 codons were analyzed (Supplementary Data 2).

## Nodule sections

For analysing the subcellular localization of SYMREM1 and the LactC2 biosensor in WT and *ipd3* (*sym1-TE7*) mutant plants, the corresponding constructs were used to transform Medicago plants by hairy root transformation as described above. Nodules were harvested 2 weeks after being inoculated with *S. meliloti* (mCherry) in open pots and directly embedded in 7% low melting agarose. Semi-thin (70 µm) longitudinal sections were obtained using a vibratome microtome (VT1000S, Leica) and the sections were analyzed using a confocal microscope (Leica TCS SP8).

## Construct design

Constructs used for SYMREM1 localization assays, protoplast analyses, and yeast transformation are described in ref. [36]. The PS reporter (2x LactC2 domain), the plasma membrane marker (LTI6b), the *SYMREM1* coding sequence and all the chimeric sequences were synthesized by Life Technologies, then cloned to expression vector by Golden Gate cloning[85]. For SYMREM1 expression and purification, the *SYMREM1* coding sequence of *Medicago truncatula* was recombined into the Gateway (GW) compatible pDEST17 vector via LR-reaction. A pET303-EGFP-10xHis empty vector (provided by Dr. Nicole Gensch, University of Freiburg) was used for expression and purification of recombinant SYMREM1 (details see below). For AtREM2.1, the coding sequence template was provided by Julien Gronnier (University of Tübingen), then amplified and cloned into the expression vector by Golden Gate cloning[86]. Other AtREM expression vectors were built in a previous study[36]. All the used constructs and all primers used are listed in Supplementary Data 3 and Supplementary Data 4, respectively. The sequence data from this article can be found in phytozome (https://phytozome.jgi.doe.gov/) with gene IDs: *SYMREM1* (Medtr8g097320), *SYFO1* (Medtr5g036540). All gene IDs are listed in Supplementary Data 3.

For promoter activation studies, a sequence 1 kb upstream of the start codon of the *Medicago truncatula REM2.1* gene was amplified via PCR from genomic DNA (A17) and cloned into a pENTR/D-TOPO vector. The *ProMtREM2.1*:GUS construct was created by a LR-reaction of pKGWFS7-vector and pENTR/D-TOPO: *ProREM2.1*.

## Histochemical promoter analysis (GUS-staining), WGA staining, and microscopy

The activation patterns of the *MtREM2.1* promoter were analyzed via β-Glucuronidase (GUS) activity. Transgenic roots were stained in GUS-staining solution (0.1 M $NaPO_4$; 1 mM EDTA; 1 mM $K_3Fe (CN)_6$; 1 mM $K_4Fe (CN)_6$; 1% Triton-X 100;1 mM X-Gluc) at 37 °C for 4 h in the dark. For fluorescent visualization of fungal structures, colonized roots were fixed in 50% ethanol for at least 12 h and afterwards cleared for 2 days at room temperature in 10% KOH. After a washing step with distilled water, roots were incubated in 0.1 M HCl for 1 h at RT. Prior to the final staining, roots were washed with distilled water and rinsed once with 1× PBS (phosphate buffered saline; pH7.4). Roots were placed in a 1× PBS-WGA–AlexaFluor594 staining solution (0.2 µg/mL WGA-AlexaFluor594; Thermo Fisher Scientific, Germany) for at least 6 h at 4 °C in dark.

## Expression analysis

Total RNA extraction was performed according to the Spectrum Plant Total RNA Kit (Sigma-Aldrich, Germany) manual. Root material was grinded in liquid nitrogen and 100 mg per root sample was used for extraction. Extracted RNA was treated with DNase I, Amp Grade (Invitrogen, Germany). The absence of genomic DNA was verified via PCR. Synthesis of cDNA was performed with 700 ng of RNA in a total reaction volume of 20 µl using the Superscript III kit (Invitrogen, Germany). For qRT-PCR analysis, a Fast SYBR Green Master Mix (Applied Biosystems, Germany) was used in a 10 µl reaction volume. A CFX96TM Real-Time system (Bio-Rad, Germany) was used for PCR reactions and detection. Expression was normalized to *Ubiquitin*. At least three biological replicates were analyzed in technical duplicates per treatment.

## Transformation of *Nicotiana benthamiana* leaves and protoplast isolation

Transgenic *Agrobacterium tumefaciens* carrying the plasmids of interest were grown in LB liquid culture overnight at 28 °C with the appropriate antibiotics. The culture was centrifuged (4000 rpm, 2 min) and the pellet was resuspended in Agromix (10 mM MgCl₂; 10 mM MES/KOH pH 5.6; 150 μM Acetosyringone) to an $OD_{600}$ of 0.3. Bacteria were mixed with the silencing suppressor p19 before being incubated for 2 h at 25 °C in darkness and then infiltrated at the lower site of *N. benthamiana* leaves. Two days after infiltration, the transformed leaves were harvested to isolate protoplasts according as described earlier[87] with a small modification (PNT solution contained 300 mg/l of CaCl₂). All experiments using protoplasts were done at least three times independently.

## Microcapillary assay

Isolated protoplasts were embedded in 0.5% agarose on a cover of a Petri Dish. The injection set-up consisted of an inverted microscope (Zeiss Axiovert 135 TV) with a motor driven micromanipulator (LANG GmbH & Co. KG, Type: STM3) mounted at the right side of the stage. Femtotips injection needles (Eppendorf) were adapted by removing the sharp-pointed tip of the needle by hand, until obtaining a needle that could not penetrate the protoplast plasma-membrane.

## Confocal laser-scanning microscopy

Protoplast images were obtained using a Leica TCS SP8 confocal microscope equipped with a HC PL APO 20x/0.75 IMM CORR CS2 objective (Leica Microsystems, Mannheim, Germany), with the exception of images in Supplementary Fig. 7b and Fig. 5a that were taken with a Zeiss LSM880 Airyscan using a 63x/1.4 oil immersion lens. GFP was excited with a White Light Laser (WLL) at 488 nm and the emission detected at 500–550 nm. YFP was excited with a 514 nm laser line and detected at 520–555 nm. mCherry fluorescence was excited at 561 nm and emission was detected between 575–630 nm. Samples, co-expressing two fluorophores were imaged in sequential mode between frames. All images were taken as z-stacks (internal distance is 0.5 um) and further analyses and projections were performed with either ImageJ/(Fiji) software[88] or Imaris.

For GUV experiments: Images were acquired by confocal fluorescence microscopy (Nikon Eclipse Ti-E inverted microscope using a Nikon A1R confocal laser scanning system with laser lines: 405 nm, 488 nm, 561 nm, 640 nm; 60x/1.49 oil immersion objective; Nikon Instruments, Inc.) and analyzed using the corresponding NIS Elements software (NIS Elements Confocal 5.20, Nikon Instruments, Inc.) and ImageJ (Fiji win64, Open source).

## Super resolution microscopy

Super resolution images were generated using an ELYRA7, structured-illumination-microscope with 3D Lattice SIM (Zeiss), equipped with a 63x/1.4 Oil DIC objective, a Pecon incubation chamber ensuring stable temperature and processed using the Zen black software package (Zeiss). Samples were stained as indicated and imaged as z-stack with 0,091 μm interval. Each stack was processed as single plane or maximum intensity projection (MIP) using Imaris software 64×9.9.0. Brightness and contrast were applied equally to all images prior to quantification. For colocalization analysis, we performed 2D line scan analysis on single plane and MIP images as indicated using Zen black software and Imaris.

## Protein expression and purification

*E. coli* BL21(DE3) cells were transformed with the plasmid pDEST17 encoding His-SYMREM1 (lipid strips, electron microscopy and actin co-sedimentation assay) and His-GFP-SYMREM1 (GUVs assembled with negatively charged lipids) proteins and the plasmid pET303-EGFP-10xHis encoding SYMREM1-GFP-His, SYMREM1^IDR-GFP-His and

SYMREM1^C-term-GFP-His (GUVs assembled with Ni-NTA lipids). A single colony of transformed *E. coli* was transferred into LB medium and grown overnight for obtaining a pre-culture. Then, 40 ml of pre-culture was inoculated in 2 l of LB media and further grown at 37 °C. Protein expression was induced by 1 mM IPTG at an $OD_{600}$ of 0.5–0.6. Afterwards, cells were incubated overnight (about 20 h) at 25 °C. Cells were harvested by centrifugation at $6000 \times g$ for 15 min. The cell pellet was resuspended in 100–150 ml Lysis buffer (20 mM HEPES, 500 mM NaCl, 20 mM imidazole, 10% glycerol, 1 mM EDTA, 1 mM Pefabloc, pH 7.2) and cells were passed through Constant Cell Disrupter (Constant Systems Limited). Cell debris was removed by centrifugation at $120,000 \times g$ for 45 min. The cleared cell lysate was loaded onto IMAC columns (5 ml HisTrap_FF) pre-equilibrated with Loading buffer A (20 mM HEPES, 500 mM NaCl, 20 mM imidazole, pH 7.2) and washed with 10 column volumes (CV) of Loading buffer A. Proteins were eluted with a linear gradient of imidazole from 20 to 450 mM in 15 CV. The eluted fractions were pooled and concentrated by spin filtration to 5 ml. Precipitated proteins were removed by an additional centrifugation for 10 min at $10,000 \times g$ before loading onto gel-filtration column (HiLoad 16/60 Superdex 200 pg) equilibrated with PBS. Eluted fractions after gel-filtration were analyzed with SDS-PAGE. Those fractions containing pure His-SYMREM1 were pooled and concentrated by spin filtration to the working concentration. For Immunoblot Analysis: 2 weeks old nodules were harvested. Extracted proteins were separated on a 12% (w/v) SDS-PAGE gel and transferred overnight at 30 V to a PVDF (polyvinylidene difluoride) membrane. Membranes were then blocked with 5% (w/v) milk for 1 hour at room temperature before being hybridized with the SYMREM1 peptide antibodies[35] at a dilution of 1:500 for 1.5 h at room temperature. Membranes were then washed with TBST three times for 10 min before incubating with a second antibody (anti-rabbit (Sigma), 1:2000 dilution, 1 h at room temperature). Prior to image acquisition, the membranes were washed again with TBST three time for 10 min.

mDia1-ct proteins were purified from the *Escherichia coli* strain BL21 CP using the protocol for C-terminal constructs (mDia1-ct) as previously described[89].

## Preparation of giant unilamellar vesicles (GUVs)

1,2-dioleoyl-*sn*-glycero-3-phosphocholine (DOPC), cholesterol and 1,2-dioleoyl-sn-glycero- 3 - [(N- (5 - amino − 1 - carboxypentyl) iminodiacetic acid) succinyl] (18:1 DGS-NTA(Ni); abbreviated Ni-NTA) were obtained from Avanti Polar Lipids (Alabaster, AL, USA). Atto 647 N 1,2-dioleoyl-*sn*-glycero-3-phosphoethanolamine (Atto 647N-DOPE) was purchased from Sigma-Aldrich Chemie GmbH (Darmstadt, Germany). The negatively charged lipids, brain PI(4,5)P₂ (L-α-phosphatidylinositol-4,5-bisphosphate) from porcine, 18:1 PI(3,5)P₂ (1,2-dioleoyl-sn-glycero-3-phospho-(1′-myo-inositol-3′,5′-bisphosphate)), 18:1 PI(4)P (1,2-dioleoyl-sn-glycero-3-phospho-(1′-myo-inositol-4′-phosphate)) and Soy PI (L-α-phosphatidylinositol)) were purchased from Sigma-Aldrich.

As previously described[90] GUVs were formed by a classical electroformation protocol. In brief, solutions of lipids (at a concentration of 0.5 mg/mL), composed of DOPC, cholesterol, Atto 647N-DOPE (as membrane marker), and either the NiNTA or desired negatively charged lipid of choice (at 59.7:30:0.3:10 mol%, respectively), were prepared in chloroform and spread on indium tin oxide (ITO-) covered glass slides. For negatively charged lipids, lipid solutions were pre-heated to 50 °C before spreading on ITO slides.

To remove residual solvent, the slides were incubated under vacuum for at least one hour, for the negatively charged lipids, 50 °C temperature was applied. A chamber was assembled with two slides, filled with 318 mOsm L⁻¹ sucrose solution as formation buffer, and an AC electrical field with a voltage of 1 V mm⁻¹ was applied (to the chamber) for 2 h at room temperature for Ni-NTA solutions, while for negatively charged lipids this was done at 50 °C temperature, to ensure a homogeneous distribution of lipids[91]. GUVs

were observed in hand-made chambers[90] using 318 mOsm L$^{-1}$ PBS as imaging buffer.

## Actin co-sedimentation assay

Purified SYMREM1 (10 μM or 30 μM) or mDia1-ct (10 μM) proteins were incubated with 10 μM G-actin (Cytoskeleton Inc.) for 1 h at RT in an actin-polymerization buffer (F-buffer: 5 mM Tris-HCl pH 7.5, 100 mM KCl, 1 mM MgCl$_2$, 0.2 mM CaCl$_2$, 0.2 mM EGTA, 0.2 mM ATP and 0.5 mM DTT). After ultracentrifugation at 80,000 × g for 30 min the supernatant was removed, as the non-pelleted fraction. The remaining pellet was then washed one time with F-buffer and subsequently resuspended in an equal volume compared to the supernatant. Laemmli buffer was added to all lysates, boiled at 96 °C for 10 min and conducted to SDS-page analysis.

## PIP lipid strips

The recombinant His-SYMREM1 protein was used for lipid binding assays as previously described[92].

## Phalloidin staining and SYMREM1 immunolocalization

Nodules (10 dpi) of *M. truncatula* were harvested in fixative solution (4% paraformaldehyde, 0.1% Triton X-100 and 10% DMSO in ASB solution (100 mM PIPES pH = 6.8, 1 mM MgCl$_2$, 1 mM CaCl$_2$)). Samples were then twice incubated under vacuum for 15 min and kept at room temperature for 4–6 h. Nodules were then embedded in 7% low melting agarose, sectioned into semi-thin (70 μm) longitudinal sections using a vibratome (VT1000S, Leica) and subsequently kept in a fixative solution for an additional hour at room temperature. After fixation, nodules sections were washed with ASB twice and kept in blocking solution (ASB supplied with 3% bovine serum albumin (BSA), 0.2% cold fish gelatin) for 30 min (room temperature). After blocking, nodule sections were incubated with ASB buffer additionally supplied with the SYMREM1 antibody (1:250) overnight. After incubation, sections were washed with ASB 3–5 times for 5 min before incubating with goat anti-rabbit Alexa488 (Molecular Probes) secondary antibodies (1:500 in ASB) and Phalloidin (Alexa Fluor™ 568) for at least 2 h at room temperature. Samples were then finally washed 3–5 times with ASB prior to imaging.

## Electron microscopy

Transmission electron microscopy (TEM) was performed on nodules harvested at (3 wpi). Nodules were cut longitudinally in half, immediately fixed in MTSB buffer[93] containing 2.5% glutaraldehyde and 4% p-Formaldehyde under vacuum for 15 min (twice), and stored at 4 °C in fixative solution until used further. After washing five times for 10 min each with buffer, nodules were post-fixed with 1% OsO$_4$ in H$_2$O at 4 °C for 2 h and again washed five times (10 min each) with H$_2$O at room temperature. The tissue was in block stained with 1% Uranyl Acetate for 1 hour in darkness, washed three times (10 min each) in H$_2$O, and dehydrated in EtOH/H$_2$O graded series (30%, 50%, 70%, 80%, 90%, 95% 15 min each). Final dehydration was achieved by incubating the samples twice in absolute EtOH (30 min each) followed by incubation in dehydrated acetone twice (30 min each). Embedding of the samples was performed by gradually infiltrating them with Epoxy resin (Agar 100) mixed with acetone at 1:3, 1:1 and 3:1 ratio for 12 h each, and finally in pure Epoxy resin for 48 h with resin changes every 12 h. Polymerization was carried out at 60 °C for 48 h. Ultrathin sections of ~70 nm were obtained with a Reichert-Jung ultra-microtome and collected in TEM slot grids. Images were acquired with a Philips CM 10 transmission electron microscope coupled to a Gatan BioScan 792 CCD camera at 80 kV acceleration voltage.

Scanning Electron Microscopy was carried out on freshly isolated protoplasts and on longitudinal vibratome sections (70 μm) of nodules collected after 3 wpi. The material was immediately fixed, dehydrated in graded EtOH series until 100% EtOH, and critical point dried in absolute EtOH·CO$_2$. Dried material was mounted on carbon tabs and coated with 5 nm platinum. Imaging of samples was performed using a Hitachi S-4800 microscope at 5 kV acceleration voltage.

Negative staining of purified SYMREM1 protein was performed by applying 5 μl protein solution to glow-discharged 400 Cu mesh carbon grids for 10 min, blotting and negatively staining using 2% (w/v) uranyl acetate. Images were recorded under low-dose conditions on a Talos F200C transmission electron microscope operated at 200 kV and equipped with a Ceta 16 M camera. Micrographs were taken at a nominal magnification of 73,000x. A total of 389 segments were manually selected using RELION-3.1.0[94]. The defocus and astigmatism of the images were determined with CTFFIND4.1[95] and numerical phase-flipping was done to correct for effects of the contrast transfer function using RELION-3.1.0. Image processing was done using IMAGIC-5[96]. Particle images were band pass filtered between 400 and 10 Å, normalized and centered by iteratively aligning them to a vertically oriented class average. Class averages containing 5–10 images were obtained by four rounds of classification based on multivariate statistical analysis, followed by multi-reference alignment using homogenous classes as new references.

## Protein 3D prediction

In-silico structure predictions were made with DeepMind AlphaFold 2.1.1 in multimer mode[44,45]. Electrostatic surface potentials were generated with ABPS[97], figures were created with PyMOL (Schrödinger LLC).

## Reporting summary

Further information on research design is available in the Nature Portfolio Reporting Summary linked to this article.

## Data availability

All data are available in the main text or the supplementary materials. Source Data are provided Source data are provided with this paper.

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

## Acknowledgements

We would like to thank Carmen Schubert and Rosula Hinnenberg for their excellent technical help and the entire Ott lab team for fruitful

discussions and providing their individual expertise throughout the course of the project. We also thank the staff of the Life Imaging Center (LIC) in the Hilde Mangold House (HMH) of the Albert-Ludwigs-University of Freiburg for help with their confocal microscopy resources, and the excellent support in image recording. Special thanks also to Norbert Roos and Jens Wohlmann at the Electron Microscopy Facility, Department of Biosciences, University of Oslo, Norway, for helping with the SEM sample preparation and imaging and to Falk Tauber (University of Freiburg, Cluster of Excellence livMatS) for designing and printing a tentative indentation support. Many thanks also to Nicole Gensch (BIOSS, University of Freiburg) for providing the pET303-EGFP-10xHis vector, Maria Harrison (Boyce Thompson Institute, Ithaca, USA) for providing the Medicago ABD-mCherry line and Julien Gronnier (University of Tübingen) for providing the AtREM2.1 CDS and critically reading the manuscript. J.K., C.L., and P.M.D. belong to the LRSV, which is part of the TULIP LABEX (ANR-10-LABX-41). The microscopes are operated by the Microscopy and Image Analysis Platform (MIAP) and the Life Imaging Center (LIC), Freiburg. Engineering Nitrogen Symbiosis for Africa (ENSA) project currently supported through a grant to the University of Cambridge by the Bill & Melinda Gates Foundation (OPP1172165) and UK government's Department for International Development (DFID) (T.O., P.M.D.). Deutsche Forschungsgemeinschaft (DFG, German Research Foundation) 431626755 (T.O.), 442219341 (P.W.). DFG under Germany's Excellence Strategy grant CIBSS – EXC-2189 – Project ID 39093984 (T.O./C.H./W.W./R.G./W.R.). China Scholarship Council (CSC) grants 201708080016 and 201506350004 (C.S./P.L.). DFG project number 414136422 (CLSM; T.O.), DFG project number 426849454 (TEM; T.O.) and DFG project number 406260942 (cryo-TEM; P.W.). European Union's Horizon 2020 Research and Innovation Program under the Marie Skłodowska-Curie grant agreement synBIOcarb (No. 814029; W.R.). Ministry for Science, Research and Arts of the State of Baden-Württemberg (Az: 33-7532.20; W.R.)

## Author contributions

Conceptualization, C.S., W.R. and T.O.; Investigation, C.S., M.R.-F., B.L., N.N., C.H.-R., P.L., E.S., E.V.M., N.M.G., J. Kn, H.W., L.S., J. Ke, C.L., A.A.M.F., K.E.G., E.M., C.P., P.W., T.S., P.M.D., O.E. and T.O.; Writing—Original Draft, C.S., M.R.-F., and T.O.; Writing—Review & Editing, C.S., M.R.-F., B.L., N.N., C.H.-R., P.L., E.S., E.V.M., N.M.G., J. Kn, H.W., L.S., J. Ke, C.L., A.A.M.F., K.E.G., E.M., C.P., C.H., W.W., P.W., T.S., P.M.D., O.E., R.G., W.R. and T.O.; Funding Acquisition, C.H., W.W., P.W., T.S., P.M.D., O.E., R.G., W.R. and T.O.; Supervision, C.H., W.W., P.W., T.S., P.M.D., O.E., R.G., W.R. and T.O.

## Funding

## Competing interests

A patent application has been submitted (U.S. Provisional Application Serial No.: 63/245662) on behalf of C.S., L.S., W.R. and T.O. concerning the use of SYMREM1 and other proteins for regulating symbiotic infections by altering membrane topologies. All other authors declare no further competing interests.
