## [Peer Review File · Nature Communications]

Stabilization of membrane topologies by proteinaceous remorin scaffoldsReviewer #1 (Remarks to the Author):

This paper studies the role of plant proteins of the remorin family, in particular SYMREM1 to stabilize tubular, wall-less, membrane structures that protrude inside the plant cells upon rhizobial infection. The authors provide cell biology evidences that protein binding and tubulation is dependent on the amphipathic and (sometimes) palmitoylated RemCA peptide at its C-terminus, while the disordered N-terminus does not influence tubulation. They also show that formin and actin polymerization are involved in the tube formation, but the precise mechanism is not discussed in the paper. They also perform electron microscopy and found many different classes of filaments, combined with the use of AlphaFold. They conclude that the proteins assemble as dimers (with some analogy with BAR-domains) and higher order oligomers forming a sheet-like structure interacting with the membrane through a large positively-charged patch. They eventually show the same tubule stabilization capability of other remorin 2 proteins that are present in plants with endosymbiotic infection processes.

Altogether, the authors propose an interesting parallel between remorin proteins in plants and BAR-domains in yeast or mammalian cells: tubules are stabilized by BIN1 (amphiphysin) muscle cells for instance. However, the origin of the force that produces these tubules inside the infected cell is not clear in this paper. Is formin and actin polymerization actively involved in the tube formation and SYMREM1 in further scaffolding? Images showing actin distribution at the scale of the cell could help. Are microtubules involved? In addition, like it has been done for all BAR-proteins, clear evidence of membrane scaffolding and tubulation by SYMREM1 could be obtained by incubating the purified proteins with liposomes and imaging them with electron microscopy. It would also clarify if negatively charged lipids (PS) are needed for protein binding.

Additionally, the analysis of the images is often too qualitative. The number of tubes is provided for the different conditions, but since this paper is about proteins and curvature, more details about the distribution of tubule diameter would be a plus. In the experiments where membrane curvature is obtained by indentation with a microcapillary, its amplitude is much weaker than upon infection. How is it possible to conclude that "data further support the role of SYMREM1 as a membrane topology scaffold"?

There is no support for the claim "These higher order oligomers can also be found in SYMREM1-stabilized membrane tubes in vivo where it may wrap around actin filaments in a helical arrangement" (lines 310-311).

Finally, the numbering of the main figures and Supplementary figures is completely disorganized. As a conclusion, I think that these new experiments and discussions are necessary for the paper to be accepted at Nature Communications.

Reviewer #2 (Remarks to the Author):

I am a great admirer of the authors' lab, and this manuscript collected a number of interesting and often striking observations. However, it is still preliminary, not so much in lacking data, but rather it is not clear which results are functionally important.

A harsher critic could categorically argue that all the main observations are merely curiosities, reflecting either phenomenon without obvious functions (e.g. Fig. 1A), SYMREM1 behavior under non-physiological conditions (Fig. 3A-B, Fig. 4), computational predictions unsupported by experimental data (Fig. 3C), or late and indirect effect of mutants whose reported defects begin early in the symbiosis (Fig. 1C, D, J). I myself believe that the authors' observations have revealed something interesting about SYMREM1 and remorins as a whole, which is captured by their one-sentence summary. However, the authors need to think hard about how to defend their claims with more physiologically relevant data, against the criticisms listed above.

I suggest that the authors begin the process of revision by building a model of SYMREM1 mode of action at the membrane during nitrogen fixing symbiosis. Then it may be clear what existing evidence supports this model, what is less important, and what additional evidence is needed. The model should also reconcile earlier reports on the role of SYMREM1 in microdomain formation.

It seems to this reviewer that one central question a model can address is how SYMREM1

maintains a particular membrane topology, one that is distinct from the default spherical-shaped membrane, where a rigid cell wall is lacking.

Existing evidence in this manuscript and earlier publications from the same group seem to support that the RemCA region of SYMREM1 causes or promotes membrane deformation, possibly through interactions with lipids, resulting in blebbing. The rest of the C-terminus can then generate a tubular structure. For the latter, actin could be key. So,

In addition to localizing SYMREM1 with actin in the ectopic tubes in protoplasts, can the authors co-localize these two proteins in or near physiological structures of the nodule?

Does SYMREM1 interact physically & directly with actin?

Can they show that the blebs caused by RemCA lack actin?

The filaments in Fig. 3A&B seems to be a distraction. Computational prediction suggests sheets, not filaments. The filaments they saw may be non-functional aggregates, which can't be ruled out at present. It will be more revealing to test what happens if SYMREM1 is mixed with actin.

The empty membrane spheres seen in nodule cells of *symrem1* mutant is also a distraction, since the defect in nodulation of this mutant begins with nod factor perception, which means the nodules are certainly abnormal. As a result the spheres can be a very indirect result of earlier defects. In addition, the authors presented no data on the nature of these spheres (e.g. using organelle markers). Are they really "empty"? Are they not vacuoles? Unless the authors propose a model or mechanism that can explain the generation of such spheres when SYMREM1 is absent, they may be a very indirect effect.

The confocal images in Fig 2 are saturated in the red channel, which makes the SYMREM1 fluorescence hard to see in detail. The authors should provide green channel-only images. The current images also show large areas of yellow, but SYMREM1 is not supposed to overlap with bacteria. Is this because the images are Z-stacks? If so, the authors should consider providing either 3-D views or a single layer, to avoid misunderstanding.

The tubes in Fig. 5A and 5A' are not completely aligned. They seem to have subtle differences in shape, especially at the bottom right. Could the authors double check? Maybe provide a bright-field image? They may seem a minor issue, but could help resolve the spatial arrangement of actin and SYMREM1 complexes. Do they seem to alternate, or exclude each other?

In summary, I believe that the authors should extensively revise the manuscript, include new data, and rethink the arrangement of existing data. Their insight is potentially very significant; it should not be muddled.

Reviewer #3 (Remarks to the Author):

In this study, Su et.al. showed remorin protein SYMREM1 functions as a membrane scaffold to participate in the formation of membrane tubes during rhizobial infections of legume root cells. By analysing the different domains within SYMREM1 as well as its homologs in other plant species, it is observed that SYMREM1 lattice shapes the curved membrane topology particularly on plasma membrane domains lack of cell wall. Overall the observations are interesting and unveil a novel molecular mechanism for the intracellular colonization of host cells driven by a membrane scaffold. Below are some comments and suggestions:

In Fig 1. The full name of the reporter and fluorescence tag should be labeled clearly in the figure, for example, LactC2-eGFP (PS) and mEosEM-MtSYMREM1. In the method, it is just mentioned confocal setting, but I think these images are not from a single layer image? If they are single-layer, it is better to show an overview with 3D mode with the "spatial" information.

Fig2 A-G, I think these should be placed in Fig 1 as introduction. In addition, a comparison between Fig 1C and Figure 2G with quantification analysis would be useful to tell the difference. Same for the Scanning Electron Microscopy analysis.

Is it possible to capture the dynamics of the membrane tubules/SYMERM1 during infection in WT/*symrem1* mutant?

For Fig3, the resolution of the images is not clearly, especially those indicated by the arrows. Based on this result, the authors could use the SYMERM1 proteins and truncations to incubate protoplasts/liposomes to see whether they would have the same effects as Fig 4.

In Fig5 C-F, how do the images were collected? does the cell pattern is based on the expressing constructs? If so, it seemed no tubulation formed in these cells, and how to explain the differences for cell morphology in Fig 4 and Fig5, both of which with different truncations of SYMREM1?

Lastly, the authors try to compare remorins with the BAR domain protein in animals. But plants also contain BAR proteins, with several studies in membrane shaping and endocytosis. Hence, it would be better to include this information.

Dear reviewers,

First of all, we would like to thank all of you for your constructive reviews. It took us quite some time to do all these new experiments, as it required to establish additional collaborations, that we now added to the manuscript. We are convinced that it was all worth the effort as it greatly improved this version.

In the following, we would like to address each comment individually.

We hope that our revision now matches your requests.

Kind regards

Thomas Ott

Reviewer #1 (Remarks to the Author):

... They also show that formin and actin polymerization are involved in the tube formation, but the precise mechanism is not discussed in the paper.

Answer: As requested, we now elaborated more on the possible mechanism. We tried to find a balance in between interpreting the data without speculating too widely. You find this on pages 13 and 14 of the manuscript file.

Altogether, the authors propose an interesting parallel between remorin proteins in plants and BAR-domains in yeast or mammalian cells: tubules are stabilized by BIN1 (amphiphysin) muscle cells for instance. However, the origin of the force that produces these tubules inside the infected cell is not clear in this paper.

Answer: We now explicitly included a sentence on our view of the forces mediating these tubes on page 13 (lines 18-21)

Is formin and actin polymerization actively involved in the tube formation and SYMREM1 in further scaffolding? Images showing actin distribution at the scale of the cell could help.

Answer: We thank you for raising this point, which certainly helped us to improve our work. Following your suggestions, we conducted time-lapse imaging of protoplasts and found that SYMREM1 initially accumulates in distinct foci to which the formin protein SYFO1 is subsequently recruited (Supplementary Fig. 8a). The additional fact that tube formation was blocked when depolymerizing the actin using Cytochalasin D (Fig. 5b,c) support a role of actin and possibly formins in tube growth.

In addition, and following the reviewer's comments we also checked the distribution of formin (SYFO1) and actin (LifeAct) when co-expressed with the RemCA peptide which mainly induces membrane blebs rather than the membrane tubes. We observed that actin was absent from those blebs although SYFO1 still localized to those membrane blebs (Supplementary Fig. 8b, c). These results indicate that the scaffolding function of a fully functional SYMREM1 is also needed in addition to actin during the growth of those membrane tubules.

Are microtubules involved?

Answer: Following the reviewer's question we analyzed membrane tube formation in the presence of Oryzalin (inhibits microtubule polymerization). Our result indicate that microtubules seem to not be key for this phenomenon as Oryzalin treatment only slightly reduced the frequency of tube formation (Fig. 5d).

In addition, like it has been done for all BAR-proteins, clear evidence of membrane scaffolding and tubulation by SYMREM1 could be obtained by incubating the purified proteins with liposomes and imaging them with electron microscopy.

Answer: Thank you for the suggestion which is in line with a comment of Reviewer #3. We now expressed and purified a His-GFP-tagged SYMREM1 protein from *E. coli* and first ran a PIP strips experiment (Supplementary Fig. 5a). Based on these results we assembled GUVs with the negative charged lipids PI4P, PI(3,5)P2 and PI(4,5)P2. Applying recombinant SYMREM1 to these GUVs revealed no binding to PI4P-GUVs, which is consistent with our PIP strips assay and served as a negative control (Supplementary Fig. 5b). When incubating the SYMREM1 protein with GUVs containing either PI(3,5)P2 or PI(4,5)P2, SYMREM1 clearly associated with these GUVs and induced membrane morphology changes at the binding site (Supplementary Fig. 5c, d).

To supplement these data, we also used Ni-NTA lipids instead of the negative charged lipids for GUV assembly to tighten the association of the SYMREM1 protein to these GUVs in vitro. Indeed, we observed more pronounced membrane topology changes with those GUVs (Fig. 3d, e). Conversely, those phenomena were observed when incubating Ni-NTA GUVs with the control protein GFP-His. It might be worth noticing that these topology changes induced by SYMREM1 were smaller than those observed when doing such assays with BAR proteins (Peter et al., 2014). The structures observed in our experiments more resembled the membrane blebs that we observed on protoplasts expressing the RemCA membrane anchor (Fig. 4h) or after treating SYMREM1-expressing protoplasts with Cytochalasin D (Fig. 5b). These data further support the hypothesis that actin is a main driver of tube elongation in protoplasts. That is also consistent with our hypothesis the organization of actin (which is missing in the GUVs assay) is required for the longer growth membrane tube.

It would also clarify if negatively charged lipids (PS) are needed for protein binding.

Answer: There are several studies showing that remorin proteins indeed can interact with negatively charged lipids (Perraki et al., 2012, Gronnier et al., 2017). In our revised version, we included a PIP strip binding assay. The results indicate that SYMREM1 can bind to several lipids such as PI3P, PI(4,5)P2, and PI(3,5)P2. PS did not show a positive signal in our experiments (Supplementary Fig. 5a).

Additionally, the analysis of the images is often too qualitative. The number of tubes is provided for the different conditions, but since this paper is about proteins and curvature, more details about the distribution of tubule diameter would be a plus.

Answer: In our revised version, we measured the diameter and length of those tubular structures. In addition, we also included some different morphologies of those membrane tubes (Supplementary Fig. 7c,d).

In the experiments where membrane curvature is obtained by indentation with a microcapillary, its amplitude is much weaker than upon infection. How is it possible to conclude that "data further support the role of SYMREM1 as a membrane topology scaffold"?

Answer: This is indeed an intriguing point, as SYMREM1 seems to act very different than BAR proteins, which mediate only one bending direction. When modelling the helical superstructure (Fig. 2g) we also noted that there is a great degree of flexibility of the structure, which supports these experimental findings. However, at this point, we cannot be certain about the precise alignment of SYMREM1 dimers and thus the higher order structure. Thus, we replaced this statement (page 10, lines 24-25).

There is no support for the claim "These higher order oligomers can also be found in SYMREM1-stabilized membrane tubes in vivo where it may wrap around actin filaments in a helical arrangement" (lines 310-311).

Answer: Thank you for the comment. We indeed did not want to refer primarily to our own data but to another study (Wei et al., 2020) which supports for this statement was included in our revised version. Thus, we moved this aspect into the discussion section.

Finally, the numbering of the main figures and Supplementary figures is completely disorganized.

Answer: We deeply apologize for this. It must have happened during the different iterations prior to the primary submission. We should have noticed this. We now took great care that everything is correctly referenced.

Reviewer #2 (Remarks to the Author):

I am a great admirer of the authors' lab, and this manuscript collected a number of interesting and often striking observations. However, it is still preliminary, not so much in lacking data, but rather it is not clear which results are functionally important. A harsher critic could categorically argue that all the main observations are merely curiosities, reflecting either phenomenon without obvious functions (e.g. Fig. 1A), SYMREM1 behavior under non-physiological conditions (Fig. 3A-B, Fig. 4), computational predictions unsupported by experimental data (Fig. 3C), or late and indirect effect of mutants whose reported defects begin early in the symbiosis (Fig. 1C, D, J). I myself believe that the authors' observations have revealed something interesting about SYMREM1 and remorins as a whole, which is captured by their one-sentence summary. However, the authors need to think hard about how to defend their claims with more physiologically relevant data, against the criticisms listed above.

I suggest that the authors begin the process of revision by building a model of SYMREM1 mode of action at the membrane during nitrogen fixing symbiosis. Then it may be clear what existing evidence supports this model, what is less important, and what additional evidence is needed. The model should also reconcile earlier reports on the role of SYMREM1 in microdomain formation.

Answer: Honestly, we would like to thank you for your clear words. For the friendly ones at the beginning as much as the clear ones later. We have now included many more data and hope that these are convincing to you. But let me start with some words about the functional relevance during the symbiotic interaction. We indeed have described the phenotype of *symrem1* mutant lines in this manuscript and earlier on (Lefebvre et al., 2010). Beside some more aberrant infection threads at early stages of this interaction, the main phenotype is found inside nodules and these are the sites we focus on here. The same holds true for the *ipd3* mutant that we used, which is mainly release-deficient. Compared to infection threads in root hairs, SYMREM1 protein accumulations at bacterial release sites and symbiosome membranes are way stronger. And this is where we think is one of the key differences to its functions in membrane nanodomains. As we have shown previously (Liang et al., 2018), SYMREM1 can scaffold proteins and at this stage we wonder whether this might rather be via altering membranes (e.g. by changing local lipid compositions and membrane fluidity) as suggested elsewhere in the past and just now very recently by Sebastien Mongrand's lab (<https://www.biorxiv.org/content/10.1101/2022.08.16.501454v1.full.pdf>). It should be noted here that they used the non-palmitoylated potato REM1.3. The Arabidopsis ortholog behaves different to most other REMs also in our assay, but truncation of the IDR of this protein resulted in long membrane tube formation as well (Fig. 7 a,b,i).

But, all of this is almost impossible to assess under symbiotic conditions.

It seems to this reviewer that one central question a model can address is how SYMREM1 maintains a particular membrane topology, one that is distinct from the default spherical-shaped membrane, where a rigid cell wall is lacking.

Answer: Yes, but we use this as a model to test how SYMREM1 could act on membrane invaginations (e.g. ITs and bacterial release sites) and non-cell wall containing membranes within cell wall containing cells (e.g. symbiosome membranes).

The rest of the (SYMREM1) C-terminus can then generate a tubular structure. For the latter, actin could be key. So, in addition to localizing SYMREM1 with actin in the ectopic tubes in

protoplasts, can the authors co-localize these two proteins in or near physiological structures of the nodule?

Answer: Thank you for the comment and suggestion. In our revised version, we co-localized SYMREM1 and actin around infection structure inside 10 days old nodules in 3 different ways (Fig. 6, Supplementary Fig. 10). We did that by performing co-immunofluorescence labeling using our SYMREM1 antibody and additionally stained actin with phalloidin (Supplementary Fig. 10a). We also did co-localization analysis with super-resolution microscopy using the SYMREM1 antibody on nodules of a stable transgenic Medicago line which harbors an actin reporter mCherry-ABD (Fig. 6a, e). Third, we performed hairy root transformations to express an mEosEM-SYMREM1 fusion protein and conducted a phalloidin counter stain (Supplementary Fig. 10b). All these experiments indeed showed a clear proximity of SYMREM1 and actin and are included in the manuscript.

Does SYMREM1 interact physically & directly with actin?

Answer: To understand whether SYMREM1 can directly interact with actin, we performed actin co-sedimentation assays in vitro. Results from these experiments showed that SYMREM1 does not directly interact with actin (Supplementary Fig. 9) under these conditions. However, our super-resolution microscopy analysis revealed a tight association of SYMREM1 and actin in Medicago nodule samples (Fig. 6). Taken together these data suggest either additional proteins mediating this interaction or a conformation of actin/SYMREM1 that cannot be achieved in vitro.

Can they show that the blebs caused by RemCA lack actin?

Answer: Indeed, these blebs miss are devoid of filamentous actin as now shown in the new Supplementary Fig. 8c.

The filaments in Fig. 3A&B seems to be a distraction. Computational prediction suggests sheets, not filaments. The filaments they saw may be non-functional aggregates, which can't be ruled out at present. It will be more revealing to test what happens if SYMREM1 is mixed with actin.

Answer: Thank you for the comment and suggestion. The computational prediction also suggests that SYMREM1 can form higher oligomers which might form helical filaments (Fig. 2g) structure. Our newly added data using recombinant SYMREM1 on GUVs suggest that these proteins are functional. Whether this is exactly this helical fold or not is something we cannot say for sure. But it is important for us to show that SYMREM1 has this auto-assembly capacity. Thus, we would prefer to keep this in the manuscript. We now improved the description and discussion of this aspect in our revised version.

Since our in vitro co-sedimentation assay not show the directly binding between SYMREM1 and actin (Supplementary Fig. 9), we did not further check SYMREM1 oligomerization patterns by TEM upon mixing with actin.

The empty membrane spheres seen in nodule cells of symrem1 mutant is also a distraction, since the defect in nodulation of this mutant begins with nod factor perception, which means the nodules are certainly abnormal. As a result the spheres can be a very indirect result of earlier defects. In addition, the authors presented no data on the nature of these spheres (e.g. using organelle markers). Are they really "empty"? Are they not vacuoles? Unless the authors propose a model or mechanism that can explain the generation of such spheres when SYMREM1 is absent, they may be a very indirect effect.

Answer: Here, we have to admit that we disagree with the reviewer. There are numerous nodulation mutants that have initial defects in primary infection steps but still develop at least some WT-like nodules. It is actually the case that such "leaky" mutants provide the opportunity to address the involvement of symbiosis-specific proteins in different steps of this interaction. We already reported earlier that *symrem1* mutants can still form infected nodules, but these have usually less infected cells and are impaired bacterial differentiation (Lefebvre et al., 2010) compared to wild-type. In addition, we also used the *ipd3* mutant in this study which still can

form the nodules after inoculation but although this gene is involved in early steps of infection (Horváth et al., 2011, Ovchinnikova et al., 2011). However, we never observed such empty membrane spheres in this genotype.

But to follow the concern, we first conducted MDY-64 staining on nodule sections from both wild-type and *symrem1* mutants. In both wild-type and the *symrem1* mutant, MDY-64 stained both tonoplast and symbiosome membranes (Fig. A in this letter). This is in fully agreement with published data that show that symbiosome membranes adopt a tonoplast identity at some stage (Limpens et al., 2009). In addition, we used a PI(3,5)P2 biosensor that further supports that these spheres have symbiosome membrane/late endosomal identity (Fig. B). However, when expressing our PS biosensor, we clearly show a lack of tonoplast labelling in WT and *symrem1* but labelling of the symbiosome membranes suggesting that these spheres are not vacuoles (Fig. 1, Supp. Fig. 1).

Thus, we hypothesize that incomplete bacterial release occasionally results in release of empty spheres rather than bacteria-filled symbiosomes. An according model has been added as (Supplementary Fig. 3; 'empty' = bacteria free).

Fig. A: Tonoplast staining of nodule sections. (a-f) Two weeks old nodules longitudinal sections from both wild-type (a) and *symrem1-1* (c and e) were obtained and stained with MDY-64. b, d and f are the 2D linescan at indicated region (red line) in a, c and e, respectively. Scale bars indicate 10 µm.

Fig. B: Symbiosome membranes in WT and empty membrane spheres in *symrem1-1* were visualized by expressing the phosphatidylinositol 3,5-bisphosphate (PI(3,5)P2) biosensor ML1N. PI(3,5)P2 mainly labels late endosome membranes

(Noack and Jaillais, 2020), and it has been shown that the symbiosome membrane later adopts an endosome membrane identity upon bacteroid elongation (Limpens et al., 2009). Scale bars indicate 5 μ m.

The confocal images in Fig 2 are saturated in the red channel, which makes the SYMREM1 fluorescence hard to see in detail. The authors should provide green channel-only images. The current images also show large areas of yellow, but SYMREM1 is not supposed to overlap with bacteria. Is this because the images are Z-stacks? If so, the authors should consider providing either 3-D views or a single layer, to avoid misunderstanding.

Answer: As SYMREM1 located to the symbiosome membrane it was difficult to image bacteria and SYMREM1 together. The images (now in Fig. 1d-f) were 3D projections (more details have been added to the figure legend and methods section). In this revised version, we also included single layer images (Supplementary Fig. 1g-i) to address the valid concern. In addition, we now also provide immunofluorescence labeling using a specific SYMREM1 antibody (Fig. 6).

The tubes in Fig. 5A and 5A' are not completely aligned. They seem to have subtle differences in shape, especially at the bottom right. Could the authors double check? Maybe provide a bright-field image? They may seem a minor issue, but could help resolve the spatial arrangement of actin and SYMREM1 complexes. Do they seem to alternate, or exclude each other?

Answer: Cautiously spotted. However, those images had to be recorded imaged in sequential mode between frames. The incomplete alignment of the images is thus due to the movement of the tubular structures during image acquisition. In addition, we now provide super-resolution images for SYMREM1 and actin in nodule samples, which suggests that they partially overlap (Fig. 6d,h).

Reviewer #3 (Remarks to the Author):

In Fig 1. The full name of the reporter and fluorescence tag should be labeled clearly in the figure, for example, LactC2-eGFP (PS) and mEosEM-MtSYMREM1. In the method, it is just mentioned confocal setting, but I think these images are not from a single layer image? If they are single-layer, it is better to show an overview with 3D mode with the "spatial" information.

Answer: We apologize that we have been slightly sloppy on this. Following your suggestions, we now labelled all images with more detailed information. In our revised version, we also included more detailed descriptions in the methods section.

Fig 2 A-G, I think these should be placed in Fig 1 as introduction. In addition, a comparison between Fig 1C and Figure 2G with quantification analysis would be useful to tell the difference. Same for the Scanning Electron Microscopy analysis.

Answer: We rearranged Fig.1 and Fig.2 according to your suggestion. The requested quantitative analysis is now also provided. Here, we scored the numbers for membrane spheres and symbiosome membranes not tightly associated with rhizobia in both WT and the *symrem1-1* mutant (for both confocal images and EM images). The data are now shown in Supplementary Fig. 2g-l.

Is it possible to capture the dynamics of the membrane tubules/SYMERM1 during infection in WT/symrem1 mutant?

Answer: The simple answer is "no". Unfortunately. But the structures are deeply buried in the tissue, which makes long-term live cell imaging of these structures impossible. Sorry.

For Fig3, the resolution of the images is not clearly, especially those indicated by the arrows.

Based on this result, the authors could use the SYMERM1 proteins and truncations to incubate protoplasts/liposomes to see whether they would have the same effects as Fig 4.

Answer: We replaced former Fig. 3A (now Fig. 2a), which now has higher resolution. Following your suggestion (as well as the one of Reviewer #1), we incubated the full-length SYMREM1 protein and different truncations of it with liposomes. In these experiments we observed that membrane topology changes in GUVs can be induced by both the SYMREM1 full-length and the C-terminal part alone (Supplementary Fig. 6). However, the N-terminal region alone does not have any effect. All those results are consistent with our observations from protoplasts assay. We also tried to incubate the purified protein on protoplast, but without success as recombinant SYMREM1 did not anchor to the surface of isolated protoplasts.

In Fig5 C-F, how do the images were collected? does the cell pattern is based on the expressing constructs? If so, it seemed no tubulation formed in these cells, and how to explain the differences for cell morphology in Fig 4 and Fig5, both of which with different truncations of SYMREM1?

Answer: Those images were collected with an inverted light microscope. Details can be found in the methods section. In addition, these protoplasts were immediately immobilized in low melting agarose. Both factors make it more difficult to see the tubes under these conditions. However, we always checked the protoplast prior to embedding and they were developing these protrusions. These data can now be found in (Supplementary Fig. 12). We did not observe any different morphologies of the protoplasts and tubes in these compared to other experiments.

Lastly, the authors try to compare remorins with the BAR domain protein in animals. But plants also contain BAR proteins, with several studies in membrane shaping and endocytosis. Hence, it would be better to include this information.

Answer: We have modified the discussion and added some more information in the discussion (page 13). However, since a direct comparison between BAR proteins and SYMREM1 is still speculative, we tried to not stress this aspect too much. We hope that this is ok.

Reviewer #1 (Remarks to the Author):

The authors did a great job with new data that reinforce the paper. I find it convincing and I think it will provide new insights that could interest scientists beyond plant biology. The revised version is suitable for publication in Nature Communications.

Reviewer #3 (Remarks to the Author):

The authors have addressed most of my questions. Just a few minor comments:

1. Order in the abstract does not fit the data presented. The structural information has been shown in fig 2, which however is mentioned in the last part of the abstract.
2. Data from Supplementary Fig. 6. can be placed in Fig3, and better to include a quantitative analysis.
3. The association between mCherry-SYMREM1 and actin (LifeAct or formin) upon Cytochalasin D or Oryzalin treatment would provide more clear evidence to support whether it is actin-dependent. This can be included in Fig 5 or Fig 6.
4. The last two paragraphs in the discussion also need further revision.

REVIEWERS' COMMENTS

First of all, we would like to thank the reviewers for their careful evaluation of the manuscript and their openness towards a new model of membrane scaffolding proposed here. Overall, we think that all these comments and the additional experiments significantly improved the manuscript.

Reviewer #1 (Remarks to the Author):

The authors did a great job with new data that reinforce the paper. I find it convincing and I think it will provide new insights that could interest scientists beyond plant biology. The revised version is suitable for publication in Nature Communications.

A: Thank you for supporting our manuscript.

Reviewer #3 (Remarks to the Author):

The authors have addressed most of my questions. Just a few minor comments:

1. Order in the abstract does not fit the data presented. The structural information has been shown in fig 2, which however is mentioned in the last part of the abstract.

A: We have arranged the abstract accordingly.

2. Data from Supplementary Fig. 6. can be placed in Fig3, and better to include a quantitative analysis.

A: The liposome assay with truncated SYMREM1 is placed in Figure 3 of the revised manuscript. And, quantitative analysis is now also included in the new Figure 3.

3. The association between mCherry-SYMREM1 and actin (LifeAct or formin) upon Cytochalasin D or Oryzalin treatment would provide more clear evidence to support whether it is actin-dependent. This can be included in Fig 5 or Fig 6.

A: Thanks for raising this point. As suggested, we have included the result about co-expressing mCherry-SYMREM1 and SYFO1-GFP (formin) upon Cytochalasin D treatment (see new Fig. 5) in the revised manuscript.

4. The last two paragraphs in the discussion also need further revision.

A: We appreciate this comment, but were also a bit lost as the reviewer did not precisely state what she/he would like to see changed. We have done our best to improve these last paragraphs further by adding a few additional sentences.